# Designing Injective and Low-Entropic Transformer for Short-Long Range Encoding

## Abstract

Multi-headed self-attention-based Transformers have shown promise in different learning tasks. Albeit these models exhibit significant improvement in understanding short-term and long-term contexts from sequences, encoders of Transformers and their variants fail to preserve layer-wise contextual information. Transformers usually project tokens onto a sparse manifold and fail to preserve injectivity among the token representations. In this work, we propose `TransJect`, an encoder model that guarantees a theoretical bound for layer-wise distance preservation between a pair of tokens. We propose a simple alternative to dot product attention to ensure Lipschitz continuity. This allows `TransJect` to learn injective mappings to transform token representations to different manifolds with similar topology and preserve Euclidean distance between every pair of tokens in subsequent layers. Evaluations across multiple benchmark short- and long-sequence classification tasks show maximum improvements of 6.8% and 5.9%, respectively, over the variants of Transformers. `TransJect` achieves the best average accuracy on the long-range arena benchmark, showcasing its superiority in capturing temporal and spatial hierarchical relationships from long sequences. We further highlight the shortcomings of multi-headed self-attention from the statistical physics viewpoint. Although multi-headed self-attention was incepted to learn different abstraction levels within the networks, our empirical analyses suggest that different attention heads learn randomly and unorderly. On the contrary, `TransJect` adapts a mixture of experts for regularization; these experts are found to be more orderly and balanced and learn different sparse representations from the input sequences. `TransJect` exhibits very low entropy, and therefore, can be efficiently scaled to larger depths.

## 1 Introduction

Over the past few decades, Deep Neural Networks (DNNs) have greatly improved the performance of various downstream applications. Stacking multiple layers has been proven effective in extracting features at different levels of abstraction, thereby learning more complex patterns (Brightwell et al., 1996; Poole et al., 2016). Since then, tremendous efforts have been made to build larger depth models and to make them faster (Bachlechner et al., 2020; Xiao et al.). Self-attention-based Transformer model (Vaswani et al., 2017) was proposed to parallelize the computation of longer sequences; it has achieved state-of-the-art performance in various sequence modelling tasks. Following this, numerous efforts have been made to reduce computation and make the Transformer model suitable even for longer sequences (Katharopoulos et al., 2020; Peng et al., 2020; Kitaev et al., 2020; Beltagy et al., 2020; Press et al., 2021; Choromanski et al., 2021; Tay et al., 2021). However, very few of these studies discuss information propagation in large-depth models. A recent study (Voita et al., 2019) characterized how token representations in Transformers change across layers for different training objectives. The dynamics of layer-wise learning are essential to understand different abstract levels a model could learn, preserve and forget to continue learning throughout the layers. However, recent studies need to shed more light on how Transformers preserve contextual information across different layers (Wu et al., 2020; Voita et al., 2019). Preserving distance and semantic similarity across layers is vital to ensure that the model learns continuously without forgetting any previously-learned knowledge, much like humans. To understand how Transformer encodes contextual similarities between tokens and preserves

the similarities layer-wise and across different attention heads, we highlight an example in Figure 1. We select three pairs of semantically-related tokens. We observe both the Euclidean and cosine distances of the representations learned at different layers of pre-trained BERT-base (Devlin et al., 2018) and Transformer trained on the IMDb sentiment classification task. We define the semantic similarity between tokens with the cosine distance between their corresponding representations. The trained Transformer model preserves the semantic similarity among the same tokens across layers. However, the Euclidean distance between their representations increases in the upper layers, indicating that Transformer projects the representations to different and sparse subspaces (a subspace with low density), albeit preserving the angle between them. On the other hand, pre-trained BERT demonstrates a more erratic trend in terms of layer-wise preservation of information. Additionally, we observe that the distances between different token representations vary across different attention heads at different encoder layers in a haphazard manner. The primary objective behind introducing multi-headed attention within Transformers is to learn different representations of tokens, attaining different linguistic properties. However, existing studies need to discuss elaborately how this information is spread across different attention heads and layers. This lack of understanding restricts us from formalizing the behaviors of self-attention-based Transformer models analytically.

Neuroscientists have been working for years to understand how the human brain functions and it simulates the bahaviors of physical sciences (Koch & Hepp, 2006; Vértes et al., 2012). Arguably, the human brain is more capable of 'associative learning' and 'behavioral formation', which can be attributed to the number of neurons and their inter-connectedness (synapses) rather than the size of the brain, or the number of layers through which the information propagates (Dicke & Roth, 2016). Although a fully-grown human brain can have hundreds of billions of neurons, it has been found that at a time, only a tiny fraction of neurons fire (Ahmed et al., 2020; Poo & Isaacson, 2009). This sparse firing can help the brain sustain low entropy (energy). *Entropy*, a measure that quantifies a state's randomness, is an essential tool to understand the factors behind human intelligence (Saxe et al., 2018; Keshmiri, 2020) and how the human brain operates. Similar attempts have been made to understand the bridge between

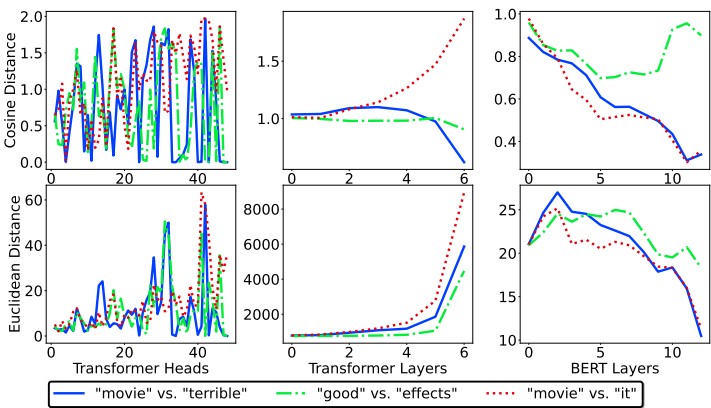

Figure 1: Layer-wise distances between a few selected tokens from the text "*This movie is terrible but it has some good effects.*" We use pre-trained BERT and Transformer models to extract token representations from different encoding layers. Both models fail to preserve the inter-token semantic distance and the spatial distance across the layers. Moreover, different attention heads across different layers learn different linguistic representations without preserving the previously-learned knowledge.

the human brain, statistical physics (thermodynamics), and information theory (Collell & Fauquet, 2015).

Unfortunately, Transformer and its variants are yet to be studied from these interdisciplinary viewpoints. The current study bridges model sparsity and entropy. Towards this, we propose a complete redesign of self-attention-based **Trans**former with enforced in**ject**ivitiy, *aka* `TransJect`. With injectivity, our model imposes the constraint in which the representations of two distinct tokens are always different across all the layers. We show that self-attention can be simplified to simple matrix multiplication with a straightforward orthogonal parameterization and rearrangement of a non-normalized attention formulation. ReZero was proposed by Bachlechner et al. (2020) to felicitate dynamical isometry and faster convergence for large-depth models. Dynamical isometry loosely means that any perturbation to the input sequence leads to a similar change to the output sequence. Networks with ReLU activation functions generally do not satisfy dynamical isometry due to the singularities (Pennington et al., 2017). As shown in Figure 1, vanilla Transformers do not propagate perturbations for all the inputs, inferring that they can also possibly not achieve dynamical isometry. With a modified formulation of ReZero, our model maintains dynamical isometry and achieves *Lipschitz* continuity, aiding in preserving layer-wise distances between tokens. By preserving layer-wise infor-

mation within a fixed bound, `TransJect` ensures incremental learning throughout the encoding layers, much similar to how humans learn continually. Kim et al. (2021) concluded that vanilla dot-product self-attention is neither Lipschitz nor injective. Injective mappings ensure reversibility, showing inherent regularization capability (Mangalam et al., 2022). Unlike Transformer, which only injects regularization through multi-headed self-attention and dropout, `TransJect` does not require explicit regularizers and can be regularized implicitly due to its inherent injectivity. To validate our hypotheses and empirically justify the superiority of our model, we use two short and five long sequence classification tasks. `TransJect` outperforms Transformer and other benchmark variants with an average margin of 3.4% and 2.2% on the short- and long-sequence classification tasks, respectively. On the long sequence classification (LRA) benchmark, our model shows the best performance, achieving 0.2% better accuracy than the best baseline, Skyformer (Chen et al., 2021). Empirical analyses suggest a very low entropy of `TransJect` representations, indicating that `TransJect` captures sparse and more orderly representations as compared to Transformer. Moreover, `TransJect` shows $13\times$ lower inference runtime than Transformer, indicating its efficiency in encoding long input sequences.[1]

## 2 Related Works

Despite being a ubiquitous topic of study across different disciplines of deep learning, Transformer (Vaswani et al., 2017) models may require better mathematical formalization. On a recent development, Vuckovic et al. (2020) formalized the inner workings of self-attention maps through the lens of measure theory and established the Lipschitz continuity of self-attention under suitable assumptions. However, the Lipschitz condition depends on the boundedness of the representation space and the Lipschitz bound of the fully-connected feed-forward layer (FFN). A similar study (Kim et al., 2021) also concluded that the dot-product self-attention is neither Lipschitz nor injective under the standard conditions. Injectivity of the transformation map is essential to ensure that the function is bijective, and therefore, reversible. Reversibility within deep neural networks has always been an active area of study (Gomez et al., 2017; Arora et al., 2015; Chang et al., 2018). A reversible network ensures better scaling in large depths and is more efficient than non-reversible structures. Recently, Mangalam et al. (2022) designed a reversible Transformer and empirically highlighted its effectiveness in several image and video classification tasks. However, any similar development has yet to be made for developing scalable reversible sequential models.

*Dynamical isometry* is a property that mandates the singular value of the input-output Jacobian matrix to be closer to one. Pennington et al. (2017) showed that dynamical isometry could aid faster convergence and efficient learning. Following their idea, Bachlechner et al. (2020) showed that residual connections in Transformers do not often satisfy dynamical isometry, leading to poor signal propagation through the models. To overcome this, they proposed residual with zero initialization (ReZero) and claimed dynamical isometry for developing faster and more efficient large-depth Transformers. Previously, Qi et al. (2020) enforced a stronger condition of *isometry* to develop deep convolution networks efficiently. However, isometric assumptions are not valid for sequential modeling due to different levels of abstraction within the input signals. Moreover, isometry may not hold between contextually-dissimilar tokens.

Another essential aspect behind designing efficient large-depth models is ensuring *model sparsity*. Several notable contributions have been made (Baykal et al., 2022; Jaszczur et al., 2021; Li et al., 2023; Tay et al., 2020a) to enforce sparse activations within Transformers to make them more efficient and scalable. Li et al. (2023) argued that sparse networks often resemble the sparse activations by the human brain, bearing a similarity between artificial and biological networks. They empirically showed that the trained Transformers are inherently sparse, and the sparsity emerges from all the layers. As discussed in the previous section, Transformers project the representations sparsely onto sparse subspaces. Although sparse models are inherently regularized and display lower entropy, by projecting the representations onto a sparse subspace, the models push themselves further from being Lipschitz, preventing them from being reversible.

Voita et al. (2019) empirically showed how information propagation across different layers of Transformer differs in different auto-regressive tasks. They concluded that Transformer forgets the contextual similarity between tokens, at the gain of better predictive performances. In this work, we show that by enforcing Lipschitz continuity, the model can preserve the contextual similarity, as well as, perform well on the final

---

[1] We commit to release the source code of `TransJect` upon acceptance of the paper.

predictive task. Our proposed `TransJect` model generates an injective mapping to project tokens onto dense subspaces, ensuring analytical Lipschitz continuity. `TransJect` learns projection function that neither squeezes nor expands signals under random perturbations. Unlike Transformers, `TransJect` projects the representations sparsely into dense subspaces. Further, it preserves the manifolds and the topology amongst the contextually-relevant tokens in every layer. With the enforced injectivity, `TransJect` establishes a theoretical guarantee for reversibility and thus can be scaled to larger depths. Further, `TransJect` displays significantly lower entropy than Transformer, indicating a lower energy footprint and more efficiency.

## 3 Background

**Activation bound.** For any function $f : \mathbb{R}^n \to \mathbb{R}^m$, we define the activation bound $K_f$ as $\sup_{\boldsymbol{x} \neq 0} \frac{||\boldsymbol{f}(\boldsymbol{x})||_p}{||\boldsymbol{x}||_p}$, for a suitable integer $p$. For a linear map $\boldsymbol{M}$, it is equivalent to the induced matrix norm $||\boldsymbol{M}||_p$. Intuitively, this is the maximum scale factor by which a mapping expands a vector $\boldsymbol{x}$. In Euclidean space, we usually choose $p = 2$.

**Lemma 1** (Activation bound of linear maps) For a matrix $\boldsymbol{M} \in \mathbb{R}^{n \times m}$, $K_{\boldsymbol{M}}$ is same as the largest absolute singular value, under $||.||_2$.

**Activation bound of self-attention.** Transformer (Vaswani et al., 2017) relies upon bidirectional multi-headed self-attention for capturing contextual similarity between tokens in the encoder. In each head, the self-attention operation projects the original representation vector $\boldsymbol{x}$ into three difference subspaces with $\boldsymbol{Q}$ (query), $\boldsymbol{K}$ (key) and $\boldsymbol{V}$ (value). Formally, Transformer calculates attention matrix $SelfAttention(\boldsymbol{X}) = Attention(\boldsymbol{X}\boldsymbol{W}^Q, \boldsymbol{X}\boldsymbol{W}^K, \boldsymbol{X}\boldsymbol{W}^V)$ as,

$$Attention(\boldsymbol{Q}, \boldsymbol{K}, \boldsymbol{V}) = \boldsymbol{D}^{-1}\boldsymbol{A}\boldsymbol{V}, \boldsymbol{A} = exp\Big(\frac{\boldsymbol{Q}\boldsymbol{K}^T}{\sqrt{d}}\Big), \boldsymbol{D} = diag(\boldsymbol{A}\boldsymbol{I}_N). \tag{1}$$

Here $d$ is the hidden dimension, and $N$ is the length of the sequence. With the normalization, $\boldsymbol{D}^{-1}\boldsymbol{A}$, Transformer ensures a stochastic attention matrix (i.e., column sum is 1).

**Corollary 1** (Largest eigenvalue of a square stochastic matrix) The largest absolute value of any eigenvalue of a square stochastic matrix is equal to 1.

As the activation bound of the matrix $\boldsymbol{D}^{-1}\boldsymbol{A}$ is 1, the activation bound of the attention map is the same as the activation bound of $\boldsymbol{W}^V$. Usually, $\boldsymbol{K}$ and $\boldsymbol{Q}$ are used to project the original representations to different subspaces. We show that having both $\boldsymbol{Q}$ and $\boldsymbol{K}$ in the same subspace with an orthogonal basis can reduce the activation bound of the attention map to singular values of $\boldsymbol{X}$.

**Lipschitz Continuity.** A function $f : \mathbb{R}^n \to \mathbb{R}^m$ under $||.||_p$ norm is called *Lipschitz continuous* if there exists a real number $K \geq 0$ such that

$$||f(\boldsymbol{x}) - f(\boldsymbol{y})||_p \leq K||\boldsymbol{x} - \boldsymbol{y}||_p. \tag{2}$$

for any $\boldsymbol{x}, \boldsymbol{y} \in \mathbb{R}^n$. Although Lipschitz continuity can be defined over any metric space, in this paper, we restrict its definition to only Euclidean space with $p = 2$. $K$ is called *Lipschitz bound*.

**Lemma 2** (Lipschitz bound for continuously differentiable functions). Any $\mathcal{C}^1$ function $f : \mathbb{R}^n \to \mathbb{R}^m$ with bounded derivative has Lipschitz bound as $sup_{\boldsymbol{x}}||\nabla_{\boldsymbol{x}} f||$.

All the proofs presented in the paper are supplied in Appendix A.

## 4 Designing Injective Transformer

This section formally describes our model, `TransJect`. It inherits the structure from the vanilla Transformer and achieves a smoother activation plane by utilizing injective maps for transforming token representations across layers. For an $L$-layered stacked encoder, we aim to learn the representation of a sequence $\boldsymbol{X} = \{\boldsymbol{x}_1, \boldsymbol{x}_2, \cdots, \boldsymbol{x}_N\}$ at each layer $l$ that preserves the pairwise distance between every pair of words within a theoretical bound. We illustrate the components of `TransJect` in Figure 2.

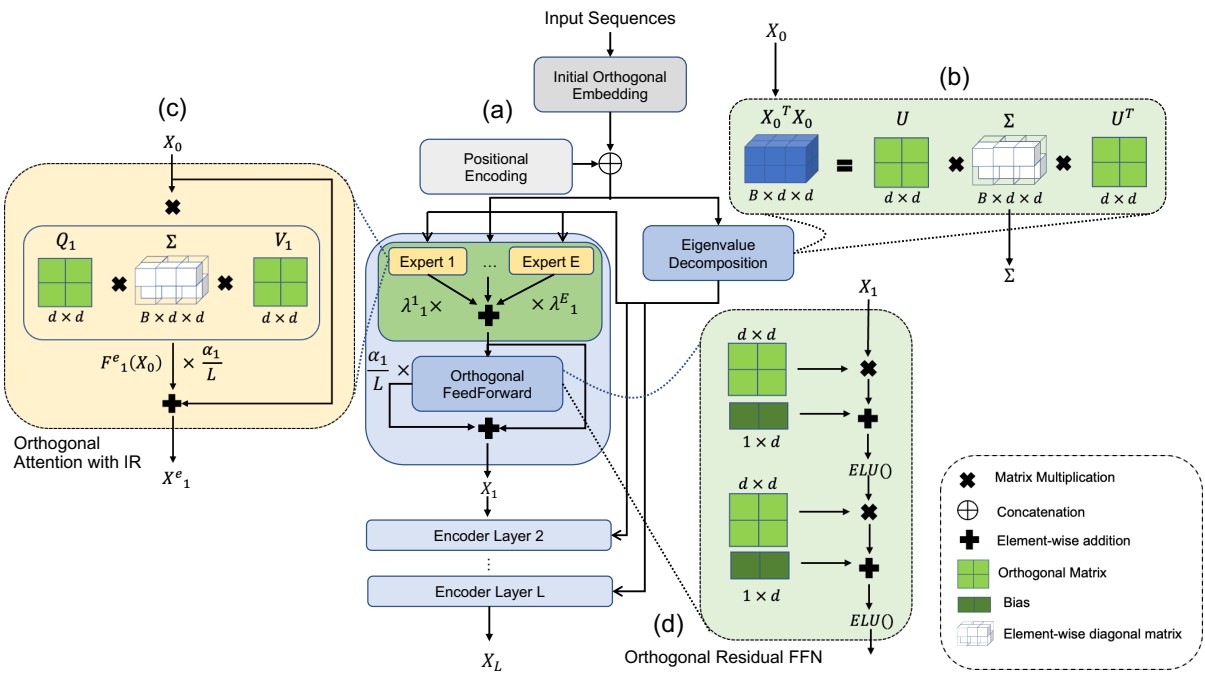

Figure 2: Internals of `TransJect` with (a) an $L$-layered encoder containing a Mixture of Experts (MOE) with $E$ experts, (b) approximated eigenvalue computation, (c) orthogonal attention with injective residual. (d) The outputs of the intermediate encoder are fed to the orthogonal residual FFN. The hidden dimension used for representing each token is denoted by $d$.

## 4.1 Space-Preserving Orthogonal Attention

The backbone of `TransJect` is the space-preserving orthogonal attention.

**Theorem 1** (Space-Preserving Orthogonal Attention) Replacing $\boldsymbol{W}^Q, \boldsymbol{W}^K, \boldsymbol{W}^V$ with real square orthogonal matrices in non-normalized linear self-attention reduces the activation bound to the largest singular value of $\boldsymbol{X}$.

*Proof.* We can compute non-normalized attention as $Attention(\boldsymbol{Q}, \boldsymbol{K}, \boldsymbol{V}) = (\boldsymbol{Q}\boldsymbol{K}^T)\boldsymbol{V} = (\boldsymbol{X}\boldsymbol{W}^Q\boldsymbol{W}^{K^T}\boldsymbol{X}^T)\boldsymbol{X}\boldsymbol{W}^V$. Here we use the linear attention (Katharopoulos et al., 2020) with the dot-product similarity between $\boldsymbol{Q}$ and $\boldsymbol{K}$. Being a real symmetric matrix, $\boldsymbol{X}^T\boldsymbol{X}$ can be decomposed into $\tilde{\boldsymbol{Q}}\boldsymbol{\Sigma}\tilde{\boldsymbol{Q}}^T$, which leads to

$$Attention(\boldsymbol{Q}, \boldsymbol{K}, \boldsymbol{V}) = \boldsymbol{X} \underbrace{\boldsymbol{W}^Q\boldsymbol{W}^{K^T}\tilde{\boldsymbol{Q}}}_{\text{orthogonal}} \underbrace{\boldsymbol{\Sigma}}_{\text{diagonal}} \underbrace{\tilde{\boldsymbol{Q}}^T\boldsymbol{W}^V}_{\text{orthogonal}}. \tag{3}$$

As the product of two orthogonal matrices is orthogonal, Equation 3 reduces to

$$Attention(\boldsymbol{Q}, \boldsymbol{K}, \boldsymbol{V}) = \boldsymbol{X}\boldsymbol{U}\boldsymbol{\Sigma}\boldsymbol{V} \tag{4}$$

with a suitable set of learnable orthogonal matrices $\boldsymbol{U}$ and $\boldsymbol{V}$, and $\boldsymbol{\Sigma}$ being the matrix containing the eigenvalues of $\boldsymbol{X}^T\boldsymbol{X}$. ∎

*Assumption:* Notice that the activation bound of the modified attention mechanism does not depend on any learnable parameters, but rather can be bounded by the largest eigenvalue of $\boldsymbol{X}^T\boldsymbol{X}$. Therefore, we assume to have a stochastic $\boldsymbol{X}^T\boldsymbol{X}$ to ensure that the largest eigenvalue is always 1, and the attention operator preserves the pairwise distance between any two tokens. In each layer, we learn orthogonal projection matrices, $\boldsymbol{U}$ and $\boldsymbol{V}$, whereas the diagonal matrix containing eigenvalues $\boldsymbol{\Sigma}$ is learned on the initial embedding obtained from the initial embedding layer defined in Section 4.4, also denoted as $l = 0$.

**Approximating eigenvalues.** Eigenvalue decomposition is computationally expensive with a runtime complexity of $\mathcal{O}(Bd^3)$, with $B$ being the batch size and $d$ being the hidden dimension. In this work, we use a simple approximation to compute $\tilde{\Sigma}$, the eigenvalues of $X^T X$. Formally, we compute $\tilde{U} = \arg\min_U ||X^T X - U\Sigma U^T||$, and $\tilde{\Sigma} = \arg\min_\Sigma ||X^T X - U\Sigma U^T||$. To learn the approximate eigenvalues, we can minimize the reconstruction loss $||X^T X - U\Sigma U^T||$ for a learnable orthogonal eigenvector $U$. We use standardization to enforce the stochasticity constraint on $\tilde{\Sigma}$. Further, instead of deriving the eigenvalues, we can also initialize a random diagonal matrix $\tilde{\Sigma}$, without any approximation that optimizes the only task-specific training objective, without enforcing the reconstruction. We denote this version of the model as `Random-TransJect`. We compute $\tilde{\Sigma}$ once, only on the initial token embeddings.

### 4.2 Injective Residual (IR)

For every layer $l$, we fine-tune the hidden representation by learning a new attention projection on the hidden state learned in the previous layer. Formally, we define,

$$X^{(l)} = X^{(l-1)} + \frac{\alpha_l}{L} F(X^{(l-1)}). \tag{5}$$

Here, $F$ is the self-attention operator, followed by a suitable non-linear activation function, and $\alpha_i \in (0, 1)$ is the residual weight. In the previous studies, ReLU and GELU (Hendrycks & Gimpel, 2016) have been popular choices for the activation function. In this work, we choose ELU (Clevert et al., 2015), a non-linear $\mathcal{C}^1$ (continuous and differentiable) activation function with a Lipschitz bound of 1. Although ReLU is a Lipschitz function with $K = 1$, it is not everywhere differentiable and injective.

Following Bachlechner et al. (2020), we adopt ReZero (residual with zero initialization) to enforce dynamical isometry and stable convergence.

**Lemma 3** (Residual contractions are injective). $f : X \to X + \frac{\alpha_l}{L} F(X)$ is injective.

To maintain the dimensionality, Transformer projects the representations to a lower-dimensional space, which reduces the total number of synapses among the neurons by a factor of $H$, the number of heads. As opposed to this, we devise a **M**ixture **of** **E**xpert (MOE) attention motivated by Shazeer et al. (2017). With this, we compute $X^{(l,e)}$ for each expert $e \in \{1, 2, \cdots, E\}$ in each layer $l$ using Equation 5, learnable expert weights $\lambda_i^{(l)}$s, and use a convex combination of them to compute,

$$X^{(l)} = \sum_{e=1}^{E} \lambda_e^{(l)} X^{(l,e)}, \qquad s.t. \sum_{e=1}^{E} \lambda_e^{(l)} = 1. \tag{6}$$

Note that $\lambda_e^{(l)}$ is computed for each sample, and the same expert weights are used for all tokens within a sample.

**Corollary 2** (Injectivity of MOE). The mapping function defined in Equation 6 is injective.

### 4.3 Orthogonal Residual FFN (ORF)

We reformulate the position-wise FFNs with orthogonal parameterization. FFN layers in Transformer emulate a key-value memory (Geva et al., 2021). We enforce Lipschitz continuity on the feed-forward sublayer to preserve the layer-wise memory. Formally, we define,

$$ORF(X^{(l)}) = X^{(l)} + \frac{\alpha_l}{L} ELU\Big(ELU(X^{(l)}W_1 + b_1)W_2 + b_2\Big). \tag{7}$$

To ensure invertibility, we use square matrices $W_1$ and $W_2$, both being orthogonally parameterized, *i.e.*, $W_1 W_1^T = W_1^T W_1 = W_2 W_2^T = W_2^T W_2 = I$.

**Corollary 3** (Injectivity of ORF). Orthogonal residual FFNs are injective.

*Proof.* Using the Lipschitz continuity of ELU, we can prove the corollary directly using Lemma 3. ∎

### 4.4 Injective Token Embedding

Positional encoding was introduced in Transformer for injecting the relative and absolute positional information of tokens into the self-attention layer. It leverages sinusoidal representations of every position and adds to the original token embeddings to infuse the positional information. To ensure injectivity at each layer, we need to ensure that the initialization of the token embeddings is also injective *i.e.*, no two tokens should have exactly the same embedding. Unfortunately, the addition operator is not injective. Therefore, we compute the initial embedding of token $x_i$ as $\boldsymbol{X}_i^{(0)} = Concat(Emb(x_i), PE_{i,:})$. Here $PE$ is defined similarly to the positional encoding proposed by Vaswani et al. (2017). Concatenation ensures the injectivity of embeddings. However, to maintain the dimensionality, we learn the initial embedding and positional encoding at a lower dimensional space, $\mathbb{R}^{\frac{d}{2}}$, where $d$ is the hidden size in the encoder.

We define the final encoder mapping for each sublayer $l$ as a composite mapping defined by,

$$SubLayer^{(l)}(\boldsymbol{X}^{(l-1)}) = ORF \circ MOE \circ IR(\boldsymbol{X}^{(l-1)}). \tag{8}$$

**Theorem 2** The composite map defined in Equation 8 is an injective Lipschitz with an upper bound of $e^2$.

A bounded activation bound ensures that the incremental learning of our encoder model reduces with large depth, which makes our model scalable to larger depths. It further enforces the importance of learning better embeddings in the initial embedding layer, which essentially drives the entire encoding. The runtime complexity of our orthogonal non-normalized attention is $\mathcal{O}(Nd^2)$, whereas dot-product self-attention has a runtime complexity of $\mathcal{O}(N^2d + Nd^2)$. In a comparable setting where $N >> d$, `TransJect` should have a lower runtime complexity than Transformer.

## 5 Experimental Setup

We evaluate `TransJect` and its variants on seven short- and long-sequence classification tasks. We use the original train-test split for all of these tasks, where the training data is used only for model training, and the test data is used for evaluating `TransJect` and the other baselines. We evaluate all the models in terms of test accuracy.

### 5.1 Tasks and Datasets

We choose the **IMDb** movie review sentiment classification (Maas et al., 2011) and the **AGnews** topic classification (Zhang et al., 2015) datasets for short text classification – the former one is a binary classification task, whereas the latter one contains four classes. For IMDb and AGnews classifications, we choose a maximum text length of 512. In all these two classification tasks, we keep the configuration of our models with $L = 6$, $E = 4$, and $d = 512$. For these

| Model | IMDb | AGnews |
|---|---|---|
| Transformer (Vaswani et al., 2017)† | 81.3 | 88.8 |
| Transformer+ReZero (Bachlechner et al., 2020) | 83.4 | 89.6 |
| Transformer+Orthogonal Parameterization | 85.1 | 86.3 |
| Linformer (Wang et al., 2020)† | 82.8 | 86.5 |
| Synthesizer (Tay et al., 2021)† | 84.6 | 89.1 |
| TransJect | **88.1** | 88.8 |
| Random-TransJect | 86.5 | **90.2** |

Table 1: Text classification accuracy on IMDb and AGnews (results highlighted with † are taken from Tay et al. (2021)).

two tasks, we use a mean pooling on the hidden representation obtained by the final encoder layer before passing it to the final classification layer. We utilize the BERT pre-trained tokenizer[2] to tokenize the texts for these two tasks.

To further highlight the effectiveness of `TransJect` on longer sequences, we evaluate our model on the LRA benchmark (Tay et al., 2020b). LRA benchmark consists of five long sequence classification tasks – ListOps (Nangia & Bowman, 2018), Byte-level text classification on IMDb review (CharIMDb) dataset (Maas et al., 2011), Byte-level document retrieval on AAN dataset (Radev et al., 2013), Pathfinder (Linsley et al., 2018),

---

[2]https://huggingface.co/bert-base-uncased

| Model | ListOps | Text | Retrieval | Pathfinder | Image | Avg. |
|---|---|---|---|---|---|---|
| Transformer (Vaswani et al., 2017)† | 38.4 | 61.9 | 80.7 | 65.2 | 40.6 | 57.4 |
| Transformer+ReZero (Bachlechner et al., 2020) | 38.3 | 59.1 | 79.2 | 68.3 | 38.6 | 56.7 |
| Transformer+Orthogonal Parameterization | 39.6 | 58.1 | 81.4 | 68.9 | 42.2 | 58.0 |
| Reformer (Kitaev et al., 2020)† | 37.7 | 62.9 | 79.0 | 66.5 | **48.9** | 59.0 |
| Big Bird (Zaheer et al., 2020)† | 39.3 | 63.9 | 80.3 | 68.7 | 43.2 | 59.1 |
| Linformer (Wang et al., 2020)† | 37.4 | 58.9 | 78.2 | 60.9 | 38.0 | 54.7 |
| Informer (Zhou et al., 2021)† | 32.5 | 62.6 | 77.6 | 57.8 | 38.1 | 53.7 |
| Nystromformer (Xiong et al., 2021)† | 38.5 | 64.8 | 80.5 | 69.5 | 41.3 | 58.9 |
| Performers (Choromanski et al., 2021)† | 38.0 | 64.2 | 80.0 | 66.3 | 41.4 | 58.0 |
| Skyformer (Chen et al., 2021)† | 38.7 | 64.7 | **82.1** | 70.7 | 40.8 | 59.4 |
| `TransJect` | **42.2** | **64.9** | 80.3 | 71.5 | 38.9 | **59.6** |
| `Random-TransJect` | 40.1 | 64.6 | 80.2 | **72.6** | 40.2 | 59.5 |

Table 2: Test accuracy on the LRA benchmark (results highlighted with † are taken from Chen et al. (2021)).

and Image classification on the CIFAR-10 dataset (Krizhevsky & Hinton, 2010). We follow the evaluation methodologies followed by Chen et al. (2021). In all these classification tasks, we keep the configuration of our models with $L = 2$, $E = 4$, and $d = 512$. Except for the text classification task, we use max pooling on the hidden representation obtained from the final encoder layer before feeding to the final classification feed-forward layer. We furnish the other hyperparameter details in Appendix B.1.

## 5.2 Results

**Short sequence classification.** Table 1 shows the performance of the competing models. On IMDb classification, `TransJect` outperforms Transformer with a 6.8% margin. `TransJect` achieves 3.5% better accuracy than Synthesizer, the best baseline. With zero residual initialization, Transformer can achieve 2.1% better accuracy in the IMDb classification task. We use an additional ablation of Transformer, where all the learnable weight matrices are parameterized orthogonally. Orthogonal parameterization improves accuracy on IMDb classification by 3.8%. On the AGnews topic classification task, `Random-TransJect` achieves 90.2% accuracy, 1.1% better than the best baseline. Interestingly, `Random-TransJect` performs better than `TransJect` on the AGnews classification task; injecting randomness through randomly initialized eigenvalues aids in 1.4% performance improvement. Limited contextual information can create difficulty reconstructing $X^T X$ from the approximate eigenvalues. Therefore, having randomly-initialized eigenvalues can aid in learning better context when the context itself is limited.

**Long sequence classification.** We evaluate `TransJect` against Transformer along with several of its recent variants and report the test accuracy in Table 2. Similar to short sequence classification tasks, `TransJect` is very effective for long sequences and consistently outperforms all the baselines. Out of five tasks, `TransJect` achieves the best performance in three and beats the best baseline, Skyformer by 0.2% on the leaderboard. `TransJect` achieves 2.3% and 0.2% better test accuracies on ListOps and byte-level text classification tasks, respectively, than the corresponding best baselines, Big Bird and Skyformer. Interestingly, `Random-TransJect` achieves the best performance on the Pathfinder task, with a wide margin of 1.9%. The ListOps task evaluates the ability to learn long-range hierarchical dependencies, whereas the Pathfinder task evaluates the ability to learn spatial dependencies. As argued by Chen et al. (2021), learning both these dimensions poses difficulty for self-attention-based methods. With a superior performance on both tasks, `TransJect` showcases its effectiveness in learning long-term patterns from both temporal sequences and sequences with different hierarchical dimensions. Moreover, the `Random-TransJect` performs better than `TransJect` on both Pathfinder and Image classification tasks with a margin of 1.1% and 1.3%, respectively. We argue that the sparsity in inputs in these two visual tasks is difficult to be approximated by the eigenvalues, which costs `TransJect` in these tasks.

## 5.3 Robustness of `TransJect` under different configurations

We evaluate `TransJect` under different hyperparameter settings to understand its robustness under different configurations. For this purpose, we choose IMDb and byte-level text classification (CharIMDb) and high-

light the test accuracy in Figure 3. On both the tasks, we observe that shallower `TransJect` is preferred over the deeper model; however, the standard deviation is a meagre 1.7% and 0.8%, respectively, compared to 2.7% and 6.6% of Transformer. Although increasing hidden size improves performance on shorter text classification, hidden size does not have any orderly impact on longer sequence classification. On the other hand, more experts in MOE typically help `TransJect` achieve better generalisation and improve performances.

## 6    Analysis

To understand the connections behind model depth, activation bounds and entropies, we conduct detailed statistical analyses on our model and Transformer. We use the IMDb and CharIMDb classification tasks for these studies as part of short long-range classification tasks, respectively. We use the outputs inferred by our models on a subsample of the test data for these analyses.

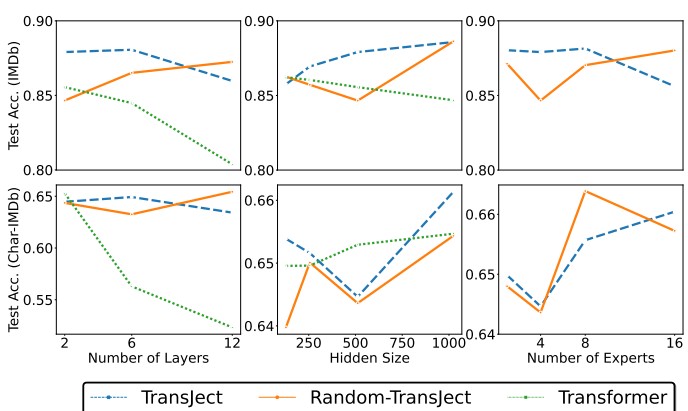

Figure 3: Performances of `TransJect` and Transformer under different configurations on the IMDb and CharIMDb classification tasks. Compared to Transformer, `TransJect` displays more stable behavior under different configurations.

**Activation bounds and entropy.** Continuing our initial discussion on preserving layer-wise distances between tokens, we calculate the distribution of activation bounds at different encoding layers. As defined in Section 3, we compute the *activation factor* for each encoding layer for `TransJect` and Transformer. Formally, for $l^{th}$ layer, we compute the activation factor

$$\mathbb{A}^{(l)} = \mathbb{E}_X \mathbb{E}_{i \neq j} \left[ \frac{||X_i^{(l)} - X_j^{(l}||}{||X_i^{(0)} - X_j^{(0)}||} \right]. \tag{9}$$

Here $X^{(l)} \in \mathbb{R}^{N \times d}$ is the hidden representation of the sequence $\boldsymbol{X}$ at $l^{th}$ layer. Similarly, we compare the *differential entropy* (*aka.* entropy) of the hidden representations learned by `TransJect` at each layer to understand how the entropy state changes across the layers. We calculate differential entropy as,

$$entropy^{(l)}\left(X^{(l)}\right) = \mathbb{E}_{j,h}\left[-\log X_{j,h}^{(l)}\right] = -\mathbb{E}_j\left[\int_{\mathbb{H}} X_{j,h} \log X_{j,h} d\boldsymbol{h}\right] \tag{10}$$

Note that $X_{j,h_i}^{(l)} \underset{h_i \neq h_j}{=} X_{j,h_j}^{(l)}$ leads the entropy to $-\infty$. Thus sparser states always have lower entropy and are more deterministic. At the same time, unorderly, random and stochastic states have higher entropy. We highlight the distribution of activation factors and entropy of token embeddings at every layer of `TransJect` and Transformer in Figure 4a. The empirical activation bound of `TransJect` is $\approx 1$, much less than the theoretical bound of $e^2 \approx 7.39$. Although we do not claim for Lipschitz continuity in the dot-product space, the empirical activation bound under dot-product is also $\approx 1$. Unlike `TransJect`, Transformer has much higher empirical activation bounds. Interestingly, Transformer aims to preserve the semantic similarity at the later layers at the expense of distance; however, `TransJect` can preserve both of them with a tighter bound, leading to a more robust representation for each token. We also highlight the activation factors of Transformer with orthogonal parameterization. Although orthogonal parameterization leads to much lower activation, it is still higher than our model.

The average entropy obtained by `TransJect` on IMDb classification is $-1.3$ with a standard deviation of 0.12. On the other hand, Transformer obtains an average entropy of 5.80 with a standard deviation of 0.13.

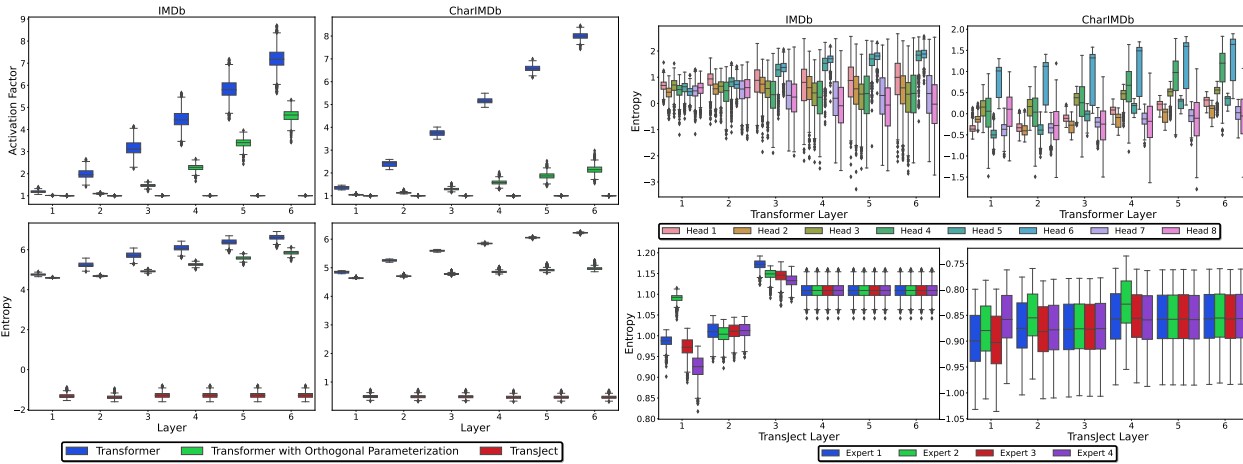

(a) Distribution of activation and entropy of models on IMDb and CharIMDb classification tasks.

(b) Attention head/expert level distribution of entropy.

Figure 4: We observe higher activation factors for Transformer, whereas the median empirical activation factor for `TransJect` is $\approx 1 << e^2$. Lower entropy for `TransJect` indicates that at the neuron level as well as at the expert level, the representations are more orderly.

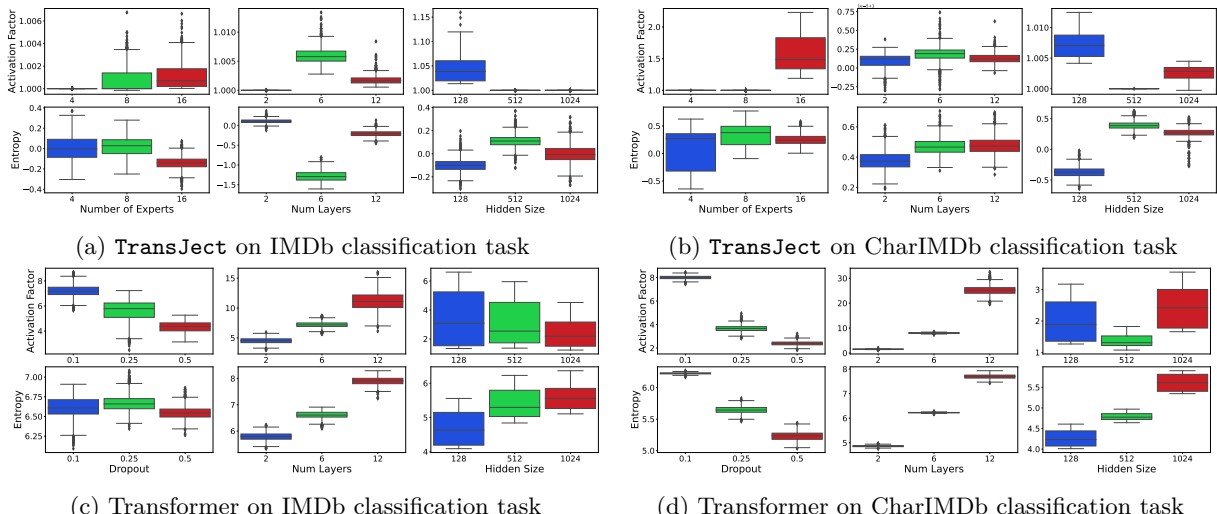

(a) `TransJect` on IMDb classification task

(b) `TransJect` on CharIMDb classification task

(c) Transformer on IMDb classification task

(d) Transformer on CharIMDb classification task

Figure 5: Distribution of activation factors and entropy with different model configurations of `TransJect` and Transformer on IMDb and CharIMDb classification tasks.

With a higher activation factor, Transformer projects the tokens onto much sparser and random subspaces, increasing the system's overall entropy. On the other hand, representations learned by `TransJect` are more orderly and thus have low entropy. Moreover, the entropy of `TransJect` does not increase perpetually, which according to the second law of thermodynamics, suggests a reversible process. We observe a high positive correlation of 0.92 between entropy and the activation factor. The same argument can be used to explain the higher entropy in later layers of Transformer. Additionally, we report the entropy of the representations at each attention head and expert level in Figure 4b. Although `TransJect` displays higher entropy at the expert level, it is still lesser than the corresponding attention heads of the Transformer model. Further, the changes in inter-quartile ranges in the later layers of Transformer show increasing randomness in larger depths, which can also be attributed to the higher activation factors. On the other hand, `TransJect` stabilises the entropy in the later layer. We also highlight the connections between activation bounds and total entropy under

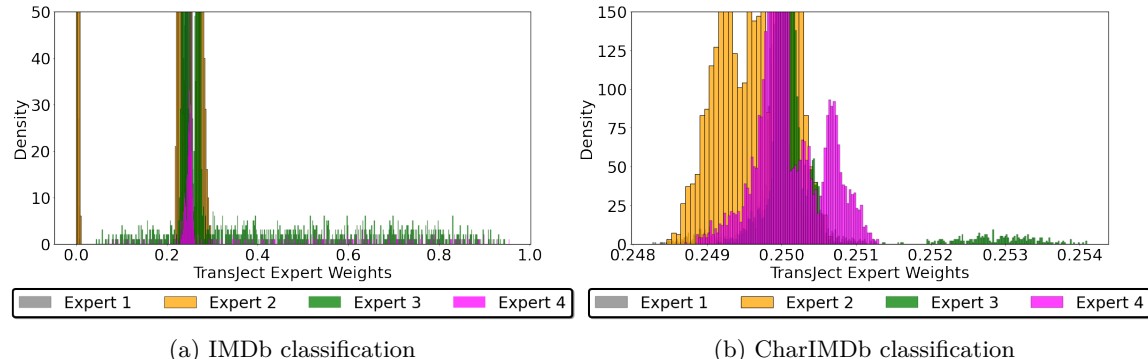

(a) IMDb classification      (b) CharIMDb classification

Figure 6: Expert weights on (a) IMDb and (b) CharIMDb classification tasks. The mode around 0.25 highlights the inherent load-balancing capabilities among the experts.

| Model | #Parameters |
|---|---|
| Transformer (Vaswani et al., 2017) | 190.1M |
| Transformer with $d_{ff} = 512$ | 95.6M |
| TransJect ($E = 1$) | 66.2M |
| TransJect ($E = 8$) | 286.8M |

(a) Comparison of number of parameters.

| Model | Speed ↑ | | | |
|---|---|---|---|---|
| | $1k$ | $2k$ | $3k$ | $4k$ |
| Transformer (Vaswani et al., 2017) | 1.0 | 1.0 | 1.0 | 1.0 |
| Transformer+Orthogonal Parameterization | 1.0 | 1.0 | 1.1 | 1.4 |
| TransJect | **5.0** | **9.6** | **12.8** | **26.3** |

(b) Comparison of inference speed.

Table 3: Comparison of the number of learnable parameters and inference speed on the CharIMDb classification task. The number of parameters is computed for encoders with $L = 6$ and $d = 512$. Transformer uses hidden size $d_{ff} = 2048$ in the feedforward layer, whereas TransJect uses the original encoder hidden dimension to maintain invertibility. We compute the test-time inference speed on long-range text classification tasks on various text lengths. The reported numbers are speedups w.r.t. Transformers.

different model configurations in Figure 5 for TransJect and Transformer, respectively. We observe that more experts in MOE increase activation at the final layer. On the other hand, with a larger dropout for Transformer, the model regularizes more and creates more sparsity, leading to lower activation factors and entropy. Contrary to TransJect, increasing the number of layers of the Transformer increase the activation. This behavior indicates the layer-agnostic property of our model.

**Effectiveness of expert model.** We observe individual importance of experts and how they interplay in the mixture model. We illustrate the expert weight distribution in Figure 6, confirming that the experts are well-balanced and that our model does not require an enforced load balancing. Further, we calculate the entropy of each expert representation to understand how it constituents the final representation at every layer. The expert entropy value of TransJect is $-2.2$. Computing an equivalent entropy of different attention heads for Transformer gives us an entropy of 0.67. The lower entropy displayed by the mixture of experts highlights the generalization capabilities of the experts. Different experts learn differently but orderly, unlike random and sparse learning by the attention heads of Transformer models.

**Preserving distances between tokens.** Figure 7 shows representations obtained on tokens of a sample text at different encoder layers, projected onto 2-D. We use isometric mapping (Tenenbaum et al., 2000) for projecting the high dimensional vectors to the $2-D$ space. TransJect maintains the initial embedding space throughout the layers, showing robustness in learning initial embeddings. On the other hand, Transformer expands the projection subspaces to more sparse subspaces, even though they project semantically similar tokens closer.

**Efficiency comparison.** We compare the models in terms of total learnable parameters in Table 3a. In an equivalent setup, TransJect has 50% more learnable parameters as compared to Transformers. We report the test-time speed on CharIMDb classification task with different length of input sequences in Table 3b. On average, we observe 13× speedup w.r.t Transformers, which increases to 26× for longer sequences of

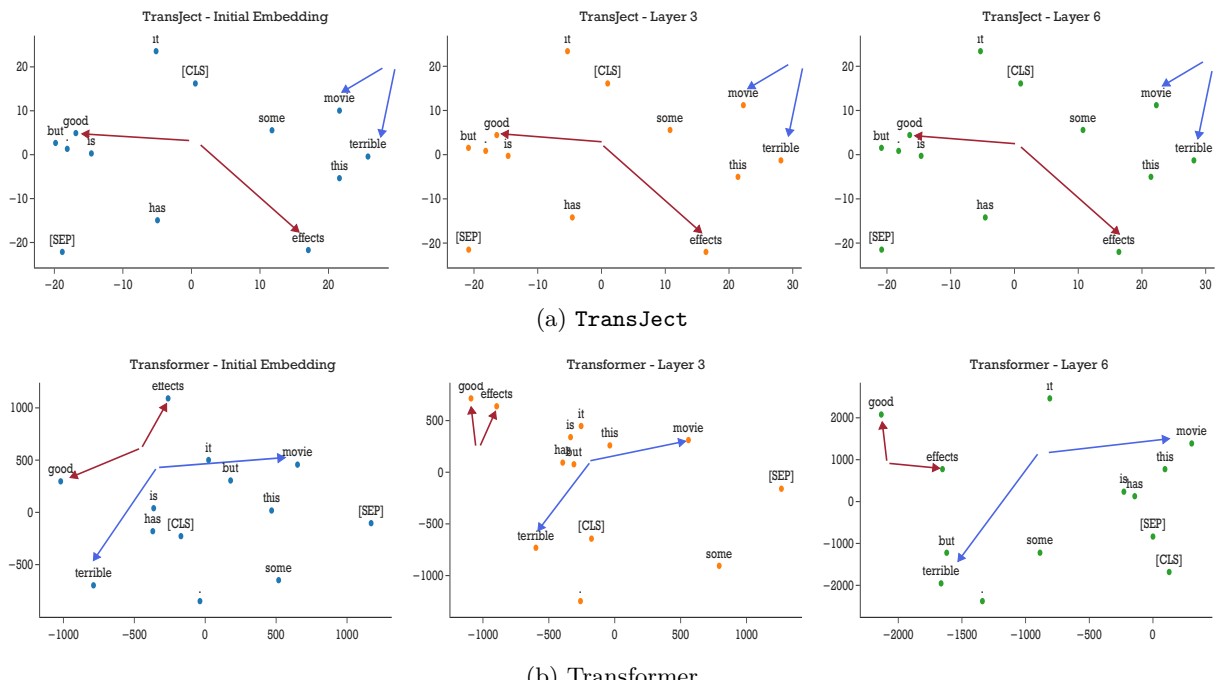

Figure 7: Isomap plot of layer-wise token embeddings learned by (a) `TransJect` and (b) Transformer on the text "*This movie is terrible, but it has some good effects.*" We highlight four semantically-relevant words and visualize their positions to understand how different models project tokens onto different subspaces in different encoding layers. `TransJect` preserves the relative distances between the tokens and their interconnectedness over the layers, indicating that the projection manifolds are topologically similar.

length more than $4k$. We further observe that orthogonal parameterization can improve efficiency for longer sequences. Albeit being highly parameterized, `TransJect` is more efficient than Transformers.

## 7   Conclusion

In this work, we introduced `TransJect`, a new learning paradigm in language understanding by enforcing a distance-preserving criterion in the multi-layered encoder models. We derived that by enforcing orthogonal parameterization and utilizing smooth activation maps, `TransJect` can preserve layer-wise information propagation within a theoretical bound, allowing the models to regularize inherently. We further argued against implicit regularization techniques like dropout, preferred injectivity, and tighter activation bounds for better generalization and efficiency. Our empirical analyses suggested a superior performance of `TransJect` over Transformers and self-attention-based baselines. Overall, we observed lower entropy with `TransJect` with low variance, which according to statistical mechanics, suggests a reversible process. Our thorough analyses also established the connection between activation bounds and entropy. These findings encourage us to explore the natural laws of science for building better, more intuitive, and more cognitive artificial intelligence. Although `TransJect` showed superior performances in several sequence classification tasks, the model assumptions may not hold for decoding. Our model relies on exact or approximated eigenvalues of the contextual vectors to encode information, an assumption that can not be applied to decoding tasks. This restricts our model from being widely used for auto-regressive tasks. Further, as explained previously, `TransJect` may be ineffective for sequences with limited context. In the future, we would like to extend these ideas to develop scalable injective auto-regressive models. Further, we will also explore the interpretability aspects of the reversible encoder and encoder-decoder models.

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

# A  Theoretical Results

We furnish the proofs of all the theoretical results presented in the main text.

## A.1  Proof of Lemma 1

$$K_M = \sup_{x \neq 0} \frac{||\boldsymbol{M}\boldsymbol{x}||_2}{||\boldsymbol{x}||_2} = \sup_{||\boldsymbol{x}||_2=1} ||\boldsymbol{M}\boldsymbol{x}||_2.$$

Squaring both the side, we decompose $\boldsymbol{M}$ as $\boldsymbol{U}\boldsymbol{\Sigma}\boldsymbol{V}^T$, where $\boldsymbol{U}$ and $\boldsymbol{V}$ are orthogonal matrices, and $\boldsymbol{\Sigma}$ is the diagonal matrix containing the singular values (square root of eigenvalues of $\boldsymbol{M}^T\boldsymbol{M}$), $\boldsymbol{\Sigma}_1 \geq \boldsymbol{\Sigma}_2 \geq \boldsymbol{\Sigma}_3 \cdots \geq 0$. For an orthogonal matrix $\boldsymbol{U}$, $||\boldsymbol{U}\boldsymbol{x}||_2^2 = ||\boldsymbol{x}^T\boldsymbol{U}^T\boldsymbol{U}\boldsymbol{x}||_2 = ||\boldsymbol{x}^T\boldsymbol{x}||_2 = ||\boldsymbol{x}||_2^2$. This leads to

$$||\boldsymbol{M}\boldsymbol{x}||_2^2 = ||\boldsymbol{U}\boldsymbol{\Sigma}\boldsymbol{V}^T\boldsymbol{x}||_2^2 = ||\boldsymbol{\Sigma}\boldsymbol{x}||_2^2 = \sum_{i=1}^n \boldsymbol{\Sigma}_i^2 x_i^2.$$

Hence,

$$K_M^2 = \sup_{\sum_{i=1}^n x_i^2=1} \sum_{i=1}^n \boldsymbol{\Sigma}_i^2 x_i^2.$$

Being a convex sum, $K_M^2 \leq \boldsymbol{\Sigma}_1^2$. Hence, $K_M \leq |\boldsymbol{\Sigma}_1|$, which completes the proof.  ∎

## A.2  Proof of Corollary 1

For any square stochastic matrix $\boldsymbol{M} \in \mathbb{R}^{n \times n}$,

$$\boldsymbol{M}\boldsymbol{I} = \begin{pmatrix} \sum_{j=1}^n \boldsymbol{M}_{j,1} \\ \sum_{j=1}^n \boldsymbol{M}_{j,2} \\ \vdots \\ \sum_{j=1}^n \boldsymbol{M}_{j,n} \end{pmatrix} = \begin{pmatrix} 1 \\ 1 \\ \vdots \\ 1 \end{pmatrix} = \boldsymbol{I}.$$

Hence, 1 is an eigenvalue of $M$. Next, we prove that 1 is the largest eigenvalue of $M$. For any eigenvalue $\lambda$ and its corresponding eigenvector $\boldsymbol{v}$, $\lambda\boldsymbol{v} = \boldsymbol{M}\boldsymbol{v}$. Without loss of generality, we assume $\arg\max_i |v_i| = 1$. Hence, for the 1st entry in this column vector $\lambda v_1 = \sum_{j=1}^n \boldsymbol{M}_{1,j} v_j$. Using triangle inequality we get,

$$|\lambda||v_1| \leq |\lambda v_1| = |\sum_{j=1}^n \boldsymbol{M}_{1,j} v_j| \leq \sum_{j=1}^n |\boldsymbol{M}_{1,j}||v_1| \leq |v_1|.$$

Hence, for any eigenvalue $|\lambda| \leq 1$. Hence, it proves our corollary that the largest eigenvalue is 1.  ∎

## A.3  Proof of Lemma 2

For any $\boldsymbol{x}, \boldsymbol{y} \in \mathbb{R}^n$, we define $g : [0,1] \to \mathbb{R}^m$ as

$$g(\boldsymbol{t}) = f(\boldsymbol{x} + \boldsymbol{t}(\boldsymbol{y} - \boldsymbol{x})). \tag{11}$$

It is easy to verify that $g(0) = f(\boldsymbol{x})$ and $g(1) = f(\boldsymbol{y})$.

$$f(\boldsymbol{y}) - f(\boldsymbol{x}) = g(1) - g(0) = \int_0^1 \nabla_{\boldsymbol{t}} g(\boldsymbol{t}) d\boldsymbol{t}. \tag{12}$$

Using chain rule of differentiation on Equation 11, we get,

$$\nabla_{\boldsymbol{t}} g(\boldsymbol{t}) = \nabla_{\boldsymbol{t}} f\Big(x + \boldsymbol{t}(y - x)\Big)(y - x).$$

Using this in Equation 12, we get

$$||f(\boldsymbol{y}) - f(\boldsymbol{x})|| = \left|\left| \int_0^1 \nabla_{\boldsymbol{t}} f\Big(\boldsymbol{x} + \boldsymbol{t}(\boldsymbol{y} - \boldsymbol{x})\Big)(\boldsymbol{y} - \boldsymbol{x}) d\boldsymbol{t} \right|\right| \leq \int_0^1 \left|\left| \nabla_{\boldsymbol{t}} f\Big(\boldsymbol{x} + \boldsymbol{t}(\boldsymbol{y} - \boldsymbol{x})\Big)(\boldsymbol{y} - \boldsymbol{x}) d\boldsymbol{t} \right|\right|.$$

As $f$ is $\mathcal{C}^1$, the supremum of its derivative exists, which allows us to set $sup_{\boldsymbol{t}} \left|\left| \nabla_{\boldsymbol{t}} f\Big(\boldsymbol{x} + \boldsymbol{t}(\boldsymbol{y} - \boldsymbol{x})\Big) \right|\right| = K$ and deduce

$$||f(\boldsymbol{y}) - f(\boldsymbol{x})|| \leq K ||\boldsymbol{y} - \boldsymbol{x}|| \int_0^1 d\boldsymbol{t} = K||\boldsymbol{y} - \boldsymbol{x}||. \quad \blacksquare$$

### A.4   Proof of Lemma 3

Let us assume $\exists \boldsymbol{x} \neq \boldsymbol{y}$ such that $f(\boldsymbol{x}) = f(\boldsymbol{y})$, which implies $||f(\boldsymbol{x}) - f(\boldsymbol{y})|| = 0$. Using triangle inequality and Equation 5 we get

$$\boldsymbol{x} - \boldsymbol{y} = f(\boldsymbol{x}) - f(\boldsymbol{y}) - \frac{\alpha_l}{L} \cdot (F(\boldsymbol{x}) - F(\boldsymbol{y}))$$

$$\implies ||\boldsymbol{x} - \boldsymbol{y}|| \leq ||f(\boldsymbol{x}) - f(\boldsymbol{y})|| + |-\frac{\alpha_l}{L}| \cdot ||F(\boldsymbol{x}) - F(\boldsymbol{y})||$$

$$\implies ||\boldsymbol{x} - \boldsymbol{y}|| \leq |\frac{\alpha_l}{L}| \cdot ||F(\boldsymbol{x}) - F(\boldsymbol{y})|| < \frac{1}{L} \cdot ||F(\boldsymbol{x}) - F(\boldsymbol{y})||$$

Here $F$ is the non-linear operator defined as $ELU(\boldsymbol{x}\boldsymbol{U}\boldsymbol{\Sigma}\boldsymbol{V})$. We can use Lemma 2 to derive the Lipschitz bound for $ELU$ non-linear function as 1 (derivative of $ELU$ is bounded by 1). Being a linear mapping, both Lipschitz bound and activation bound are the same for $\boldsymbol{U}\boldsymbol{\Sigma}\boldsymbol{V}$, *i.e.*, $||\boldsymbol{U}\boldsymbol{\Sigma}\boldsymbol{V}||$. Using the orthogonality conditions of $\boldsymbol{U}$ and $\boldsymbol{V}$ and the fact that the largest eigenvalue of $\boldsymbol{\Sigma}$ is 1, we derive the Lipschitz bound of $\boldsymbol{U}\boldsymbol{\Sigma}\boldsymbol{V}$ as 1. Using the chain rule of differentiation and properties of supremum, we calculate $sup_{\boldsymbol{x}} ||\nabla_{\boldsymbol{x}} F(\boldsymbol{x})|| \leq sup_{\boldsymbol{x}} ||\nabla_{\boldsymbol{x}} ELU(\boldsymbol{x})|| \cdot ||\boldsymbol{U}\boldsymbol{\Sigma}\boldsymbol{V}|| = 1$. Thus, the Lipschitz bound of $F$ is 1.

Using this we get

$$||\boldsymbol{x} - \boldsymbol{y}|| < \frac{1}{L} \cdot ||\boldsymbol{x} - \boldsymbol{y}||.$$

Which contradicts that fact that $\frac{1}{L} \leq 1$. Hence, we prove the lemma by contradiction. $\quad \blacksquare$

### A.5   Proof of Corollary 2

Let us assume $\boldsymbol{x}^{(l)} \neq \boldsymbol{y}^{(l)}$ such that

$$\sum_{e=1}^{E} \lambda_e \boldsymbol{x}^{(l-1)} + \frac{\alpha_l}{L} \sum_{e=1}^{E} \lambda_e ELU(\boldsymbol{x}^{(l-1)} \boldsymbol{M}^{(l-1,e)}) = \sum_{e=1}^{E} \lambda_e \boldsymbol{y}^{(l-1)} + \frac{\alpha_l}{L} \sum_{e=1}^{E} \lambda_e ELU(\boldsymbol{y}^{(l-1)} \boldsymbol{M}^{(l-1,e)}).$$

Using $\sum_{e=1}^{E} \lambda_e = 1$ we obtain,

$$\boldsymbol{x}^{(l-1)} + \frac{\alpha_l}{L} \sum_{e=1}^{E} \lambda_e ELU(\boldsymbol{x}^{(l-1)} \boldsymbol{M}^{(l-1,e)}) = \boldsymbol{y}^{(l-1)} + \frac{\alpha_l}{L} \sum_{e=1}^{E} \lambda_e ELU(\boldsymbol{y}^{(l-1)} \boldsymbol{M}^{(l-1,e)})$$

$$\implies ||\boldsymbol{x}^{(l-1)} - \boldsymbol{y}^{(l-1)}|| = ||\frac{\alpha_l}{L} \sum_{e=1}^{E} \lambda_e ELU(\boldsymbol{x}^{(l-1)} \boldsymbol{M}^{(l-1,e)}) - \frac{\alpha_l}{L} \sum_{e=1}^{E} \lambda_e ELU(\boldsymbol{y}^{(l-1)} \boldsymbol{M}^{(l-1,e)})||$$

$$\implies ||\boldsymbol{x}^{(l-1)} - \boldsymbol{y}^{(l-1)}|| < \frac{1}{L} ||\sum_{e=1}^{E} \lambda_e ELU(\boldsymbol{x}^{(l-1)} \boldsymbol{M}^{(l-1,e)}) - \sum_{e=1}^{E} \lambda_e ELU(\boldsymbol{y}^{(l-1)} \boldsymbol{M}^{(l-1,e)})||$$

Using Lipschitz bound of $ELU(\boldsymbol{x}^{(l-1)}\boldsymbol{M}^{(l-1,e)})$ as 1, we obtain

$$||\boldsymbol{x}^{(l-1)} - \boldsymbol{y}^{(l-1)}|| < \frac{1}{L}\sum_{e=1}^{E}\lambda_e||\boldsymbol{x}^{(l-1)} - \boldsymbol{y}^{(l-1)}|| = \frac{1}{L}||\boldsymbol{x}^{(l-1)} - \boldsymbol{y}^{(l-1)}||.$$

This contradicts that fact that $\frac{1}{L} \leq 1$. Hence, we prove the corollary by contradiction. ∎

## A.6 Proof of Theorem 2

Being a composition of continuously differentiable injective functions, $f = ORF \circ MOE \circ IR$ is a continuously differentiable injective function.

Next, we compute the Lipschitz bound for $f$. For the sake of simplicity, let us assume $\frac{\alpha_1}{L} = \frac{\alpha_2}{L} \cdots = \frac{\alpha_l}{L} = \alpha < \frac{1}{L}$. We first expand $f$ as

$$f(\boldsymbol{x}) = \sum_{e=1}^{E}\lambda_e\boldsymbol{x} + \sum_{e=1}^{E}\lambda_e\alpha \cdot ELU(\boldsymbol{x}\boldsymbol{U}^e\Sigma\boldsymbol{V}^e) + \alpha \cdot ELU(\overline{\boldsymbol{x}}).$$

$$\overline{\boldsymbol{x}} = ELU\Big(\sum_{e=1}^{E}\lambda_e\boldsymbol{x}\boldsymbol{W}_1 + \sum_{e=1}^{E}\lambda_e\alpha \cdot ELU(\boldsymbol{x}\boldsymbol{U}^e\Sigma\boldsymbol{V}^e)\boldsymbol{W}_1 + \boldsymbol{b}_1\Big)\boldsymbol{W}_2 + \boldsymbol{b}_2.$$

$$\implies \overline{\boldsymbol{x}} = ELU\Big(\boldsymbol{x}\boldsymbol{W}_1 + \sum_{e=1}^{E}\lambda_e\alpha \cdot ELU(\boldsymbol{x}\boldsymbol{U}^e\Sigma\boldsymbol{V}^e)\boldsymbol{W}_1 + \boldsymbol{b}_1\Big)\boldsymbol{W}_2 + \boldsymbol{b}_2. \tag{13}$$

We break down the expression and compute the Lipschitz bound of $ELU(\boldsymbol{x}\boldsymbol{U}\Sigma\boldsymbol{V})\boldsymbol{W}_1 + \boldsymbol{b}_1$ first. Using the Lipschitz continuity of ELU, we get

$$||ELU(\boldsymbol{x}\boldsymbol{U}\Sigma\boldsymbol{V})\boldsymbol{W}_1 + \boldsymbol{b}_1 - ELU(\boldsymbol{y}\boldsymbol{U}\Sigma\boldsymbol{V})\boldsymbol{W}_1 - \boldsymbol{b}_1||$$
$$= ||ELU(\boldsymbol{x}\boldsymbol{U}\Sigma\boldsymbol{V})\boldsymbol{W}_1 - ELU(\boldsymbol{y}\boldsymbol{U}\Sigma\boldsymbol{V})\boldsymbol{W}_1||$$
$$\leq ||ELU(\boldsymbol{x}\boldsymbol{U}\Sigma\boldsymbol{V}) - ELU(\boldsymbol{y}\boldsymbol{U}\Sigma\boldsymbol{V})|| \cdot ||\boldsymbol{W}_1||$$
$$\leq ||ELU(\boldsymbol{x}\boldsymbol{U}\Sigma\boldsymbol{V}) - ELU(\boldsymbol{y}\boldsymbol{U}\Sigma\boldsymbol{V})||$$
$$\leq ||\boldsymbol{x}\boldsymbol{U}\Sigma\boldsymbol{V} - \boldsymbol{y}\boldsymbol{U}\Sigma\boldsymbol{V}||$$
$$\leq ||\boldsymbol{x} - \boldsymbol{y}|| \cdot ||\boldsymbol{U}|| \cdot ||\Sigma|| \cdot ||\boldsymbol{V}||$$
$$\leq ||\boldsymbol{x} - \boldsymbol{y}||.$$

Feeding this into Equation 13 and using the fact $\sum_{e=1}^{E}\lambda_e = 1$, we get

$$||\overline{\boldsymbol{x}} - \overline{\boldsymbol{y}}||$$
$$\leq (||\boldsymbol{x} - \boldsymbol{y}|| \cdot ||\boldsymbol{W}_1|| + |\alpha| \cdot ||\boldsymbol{x} - \boldsymbol{y}||) \cdot ||\boldsymbol{W}_2||$$
$$\leq (1 + |\alpha|)||\boldsymbol{x} - \boldsymbol{y}||.$$

Finally,

$$||f(\boldsymbol{x}) - f(\boldsymbol{y})||$$
$$\leq ||\boldsymbol{x} - \boldsymbol{y}|| + |\alpha| \cdot ||\boldsymbol{x} - \boldsymbol{y}|| + |\alpha|(1 + |\alpha|) \cdot ||\boldsymbol{x} - \boldsymbol{y}||$$
$$\leq (1 + |\alpha|)^2||\boldsymbol{x} - \boldsymbol{y}||$$
$$< (1 + \frac{1}{L})^2||\boldsymbol{x} - \boldsymbol{y}||.$$

Here, $f$ is a layer-wise operator. Hence, for all the $L$ encoder layers,

$$||f(\boldsymbol{x}) - f(\boldsymbol{y})|| < (1 + \frac{1}{L})^{2L}||\boldsymbol{x} - \boldsymbol{y}||.$$

Using $\lim_{n \to \infty}(1 + \frac{1}{n})^n = e$, we get $||f(\boldsymbol{x}) - f(\boldsymbol{y})|| < e^2||\boldsymbol{x} - \boldsymbol{y}||.$ ∎

## B  Experimental Setup

### B.1  Training Setup

All IMDb and AGNews classification experiments are run for 30 epochs. We use an early stopping based on the test loss with the patience of 4 epochs to terminate learning on plateaus. For both these experiments, we use Adam optimizer with a learning rate of 0.0005, $\beta_1 = 0.9, \beta_2 = 0.98$ and $\epsilon = 10^{-9}$. We use both training and test batches of size 32. We follow the implementation details shared by Chen et al. (2021) for the LRA benchmark. All these experiments are run for $50k$ steps. One Tesla P100 and one Tesla V100 GPU are used for conducting all the experiments. For each task, we report the average accuracy across three different runs.

### B.2  Implementation Details

To enforce orthogonality during the forward pass and after backpropagation, we use PyTorch parameterization[3]. This implementation uses different orthogonal maps (e.g. householder or Cayley mapping) to ensure orthogonality. It further uses the framework of dynamic trivialization (Lezcano Casado, 2019) to ensure orthogonality after gradient descent. We use the implementation by Scikit-learn[4] that uses $1 - cosine\_similarity$ as the cosine distance. Cosine similarity is the cosine of the angle between two vectors which lies in $[-1, 1]$. Thus the distance lies in $[0, 2]$.

---

[3]https://pytorch.org/docs/stable/generated/torch.nn.utils.parametrizations.orthogonal.html
[4]https://scikit-learn.org/stable/modules/generated/sklearn.metrics.pairwise.cosine_distances.html

