# OpenReview forum: "Designing Injective and Low-Entropic Transformer for Short-Long Range Encoding"
_TMLR — Rejected by TMLR_

### Review · Reviewer_D7jj · 2023-03-29

**Summary Of Contributions:**

This paper presents a new type of Transformer architecture, primarily based on orthogonal weight parameterizations to preserve properties between different token embeddings (e.g. cosine distance).

Specifically, the following techniques are used:

* Setting the attention mechanism to have the form $XU \Sigma V$, where $U,V$ are learnable orthogonal matrices and $\Sigma$ is the eigenvalue matrix of $X^{T}X$.
* Adding residual connections after the attention output, along with orthogonal project matrices again.
* Mixture of Experts as opposed to multiheaded division.
* Concatenated positional encodings.

Experiments are conducted over Long Range Arena (LRA), and ablations are conducted to visualize token embedding distances in addition to regular performances as a function of the number of layers.

**Audience:**

Yes

**Broader Impact Concerns:**

No broader impact statements / ethical concerns required.

**Claims And Evidence:**

No

**Requested Changes:**

Please address the above weaknesses. Unfortunately, I think these are huge holes that I must currently vote "no" on satisfactory support for the paper's claims and evidence.

**Strengths And Weaknesses:**

Strengths:
 * The architecture design seems novel. Especially factoring in Figure 7, this may be one of the first Transformer architectures that preserves isometry and Lipschitz continuity between layers.
* Table 2 (LRA performance) suggests that this design does have merits in terms of overall end-to-end performance. The TransJect methods seem to at least be competitive with other models.

Weaknesses:
* Section 4.1, Theorem 1 is wrong on attention: Currently attention uses row-wise softmax on the product $QK$ (and even if we forget the normalizations, it would still be $\exp(Q K^{T})$. However, the paper claims that attention is simply $(QK^{T})V$ without any use of non-linearities over the similarity scores $Q K^{T}$, which is clearly false. It is fine if the authors wish to propose a new type of mechanism, but then it would be a huge stretch to consider this "attention". This section needs to be re-written.
    * Even under the new proposed mechanism, the runtime bounds do not clearly display the relationship against the sequence length (usually notated as $L$). This is especially required, given that Table 2 presents experimental results on LRA (which is meant to test efficient Transformers originally).
* The general hypothesis that "embeddings which preserve cosine similarities and Lipschitz continuity, lead to better performance" has some key holes:
    * The original Transformer utilizes LayerNorm normalizers, which makes it hard to describe anything about end-to-end Lipschitz continuity (in fact, LayerNorm would give an upper bound of the Lipschitz constant to be infinite). Yet, Transformers still work fine. A similar argument could be made at the very end of the Transformer, where e.g. in classification settings, softmax is used (which is a form of normalization ignores the general absolute magnitude of tensors).
    * The paper does not provide very strong evidence around this hypothesis either. For example, Figure 3 does not display a clear trend (the curves are not monotonic). Figure 1 does show somewhat of a trend between Euclidean distance and number of layers, but this is unsurprising and doesn't necessarily mean the embeddings are "bad" (especially as discussed, normalization can deal with the increasing distances).
* It is unclear how to train the model in order to preserve the orthogonality of the weights. Taking a regular gradient step may take a matrix out of the manifold of orthogonal matrices - do you project the next iterate back into the orthogonal manifold?

---

> ### Author Response · Authors · 2023-04-11
> **Author Response to Reviewer D7jj**
>
> We thank you for the valuable feedback. We address the raised concerns. We have also highlighted the changes in our manuscript.
>
> **W1.** Section 4.1, Theorem 1 is wrong on attention $\dots$
>
> **Response.**
> * We apologize for the confusion. Let us clarify this here. The non-normalized attention map in Theorem 1 is motivated by linear attention [1], where we use $QK^{T}$ to define the similarity between the query and the key. This similarity metric does not use any non-linearity to ensure that the operator is linear in terms of the input. Hence, our proposed space-preserving orthogonal attention is a simplified version of linear attention. Hope it clarifies the confusion.
>
> We have clarified this in Section 4.1 of our revised  manuscript.
>
> * In the paper, we denote the maximum sequence length with $N$ (as opposed to $L$ which is used to denote the maximum depth of the model). In Section 4.4, we already mention the runtime bounds in terms of the maximum sequence length.
>
> **W2.** The general hypothesis that ``embeddings which preserve cosine $\dots$
>
> **Response.**
> * Although normalization can increase the Lipschitz bound of the Transformer operator, Figure 4 confirms that the activation bound can be drastically reduced with an orthogonal parameterization. Orthogonal parameterization shows lower entropy and improves performance in 5 out of 7 tasks. In our paper, we highlight the relationships between activation bound, entropy and depth of the models. We conclude that Transformers with larger depths have higher activation bounds; thus, these models project token embedding into sparser subspaces, making them inefficient. We also highlight the inference speed in Table 3b on long-range classification, which shows the necessity of building efficient Transformers.
>
> * As explained in Section 5.3 of our paper, Figure 3 highlights that TransJect is more robust (lower variance) as compared to vanilla Transformer under different hyperparameter settings.
>
> * As explained above, having the embeddings in a sparser subspace is not necessarily bad. In fact, in language modeling, where the model requires learning different topological properties from texts, the model may project tokens onto different subspaces with different sparsity. However, as we argued in Section 1, sparser embeddings lead to higher entropy, which makes the model less efficient, particularly during long-sequence encoding.
>
> **W3.** It is unclear how to train the model in order to preserve $\dots$
>
> **Response.** We follow the orthogonal parameterization implementation by PyTorch [2]. This implementation is motivated by the framework of dynamic trivialization [3] to ensure orthogonality after gradient descent. We have now added this information in Section B.2 of our revised manuscript.
>
> ### References
> [1] Katharopoulos, Angelos, Apoorv Vyas, Nikolaos Pappas, and François Fleuret. *Transformers are rnns: Fast autoregressive transformers with linear attention.* In International Conference on Machine Learning, pp. 5156-5165. PMLR, 2020.
>
> [2] https://pytorch.org/docs/stable/generated/torch.nn.utils.parametrizations.orthogonal.html
>
> [3] Lezcano Casado, Mario. *Trivializations for gradient-based optimization on manifolds.* Advances in Neural Information Processing Systems 32 (2019).

---

### Review · Reviewer_1wMf · 2023-04-05

**Summary Of Contributions:**

This paper studies how transformers preserve the similarities layer-wise across layers and heads. The authors suggest that preserving such similarities helps improve the performance of the model. They then propose a new attention, namely TransJect. TransJect learns injective mappings that preserve the layer-wise Euclidean distances between every pair of tokens. Other methods, such as the Injective Residual (IR), the Orthogonal Residual FFN (ORF), and Mixture of Expert (MOE) attention, are proposed to further enhance the performance of the model. The authors then validate the advantage of TransJect on classification tasks.

**Audience:**

Yes

**Broader Impact Concerns:**

I have no concerns on the ethical implications of the paper.

**Claims And Evidence:**

No

**Requested Changes:**

**Questions for the Authors:**

1. Why can cosine distance be greater than 1 in Figure 1?

2. If transformers already preserve the cosine similarities as mentioned in the paper, why preserving the Euclidean similarities is important?

3. In Orthogonal Residual FFN, how to enforce W_1 and W_2 to be orthogonally parameterized?


**Strengths And Weaknesses:**

**Summary of the Review:**

This paper is not well organized. Many methods are proposed. However, none of them is well studied, and the proposed methods are not in a coherent scheme. Many of the proposed methods are quite similar to existing works. Furthermore, experiments are significantly lacking.

Currently, I am leaning toward rejecting the paper.

**Strong points:**

1. This paper points out an interesting phenomenon in transformers, which is that transformers do not preserve the layer-wise Euclidean similarities across layers.

2. The paper is well motivated.

**Weak points:**

1. The proposed methods are loosely related. The confusing writing and organization even make the paper harder to follow.

2. Some proposed methods are not novel. In particular, the injective residual is not new. It is similar to the skip connection in self-attention. The MOE attention is similar to the multi-head attention except for the convex combination. It is not clear to me the advantage of MOE attention over the multi-head attention. Also, for Orthogonal Residual FFN, using orthogonal weights for a feedforward neural network layer is not new (see [1]).

3. The MOE attention introduces a significant amount of additional parameters \lambda_{e}^{l}.

4. From reading the paper, it is not clear to me what attention TransJect implements? Is it given by Eqn 4?

5. Experimental results are significantly lacking. More experiments on large-scale datasets for various tasks are needed. Currently, the authors only validate their methods on small-scale datasets for classification tasks.

**References:**

[1] Xiao, L., Bahri, Y., Sohl-Dickstein, J., Schoenholz, S. &amp; Pennington, J.. Dynamical Isometry and a Mean Field Theory of CNNs: How to Train 10,000-Layer Vanilla Convolutional Neural Networks. ICML, 2018.

---

> ### Author Response · Authors · 2023-04-11
> **Author Response to Reviewer 1wMf 1/2**
>
> We thank you for the valuable feedback. We address the raised concerns. We have also highlighted the changes in our manuscript.
>
> **W1**. The proposed methods are loosely related $\dots$
>
> **Response.** It is unclear from your comments which part of the writing and organization is confusing. It would be very helpful if you precisely mention the confusion. We will be happy to edit accordingly.
>
> **W2**. Some proposed methods are not novel $\dots$
>
> **Response.**  In this paper, we propose an injective parameterization of self-attention-based Transformer models. Such changes in designing our proposed model achieve several mathematical properties that the vanilla Transformer and most of its variants currently undermine. These changes enable our model to ensure injectivity, Lipschitz continuity and dynamical isometry, making our model efficient and superior in long-sequence encoding tasks. We highlight the key differences between TransJect and the existing self-attention models below.
>
> * Firstly, with a simple orthogonal parameterization, we show that the linear attention can be brought down to simple matrix multiplication, with the Lipschitz bound of the largest singular value of $X$, the token embeddings.
>
> * We show that the residual mapping can be made injective using the properties of Lipschitz continuity. Further, using zero residual initialization, our model achieves dynamical isometry, encouraging efficiency for larger depth models. These properties are not visible with the vanilla Transformer model.
>
> * As opposed to multi-head self-attention, the mixture of experts (MoE) uses the convex sum of different attention experts, ensuring injectivity. Under standard conditions, multi-head self-attentions are neither Lipschitz nor injective.
>
> * Finally, combining all these  components, we analytically show the Lipschitz continuity of our encoder model with a theoretical upper bound. Note that the vanilla Transformer is not Lipschitz under ordinary conditions [8].
>
> **W3.** The MoE attention introducing  significant parameters $\dots$
>
> **Response.**  Usually, Transformers use multi-head self-attention, where each attention head learns different semantics. As opposed to this, we propose a mixture of experts (MoE) module using a convex combination of different expert (heads) representations. The convex sum ensures that the resultant mapping is still injective. We have now added this property in Corollary 2 of our revised manuscript and furnished the proof in Appendix A.5. Although MoE introduces $\mathcal{O}(E)$ times more parameters, as discussed in Section 4 of our paper, the proposed attention has lower runtime complexity than vanilla dot-product self-attention. Additionally, being injective, our proposed method is highly efficient in long-range encoding tasks (c.f. Table 3b).
>
> We never claim that our model is efficient in terms of  parameter size. In fact, we highlighted in Section 6 that under a comparable setting, TransJect has $50\\%$ more parameters than vanilla Transformer. Still, our model has higher computational efficiency than Transformer due to injective regularization capabilities.
>
> **W4.** Unclear what attention TransJect implements? Is it given by Eqn 4?
>
> **Response.** Yes. As described in Equation 4 of our paper, we use space-preserving orthogonal attention.
>
> **W5.** Experimental results are significantly lacking.
>
> **Response.** We respond to this comment in two parts.
>
> Regarding the concern related to the use of  small datasets, we would like to mention that the datasets we considered are fairly large (e.g., AGNews dataset has $120k$ training examples, ANN dataset $150k$ training examples, Pathfinder dataset $160k$ training examples) and are widely used to evaluate Transformer and its variants [1, 2, 3, 4, 5].
>
> Regarding the concern that we only considered the classification benchmarks, we would like to refer to a series of papers [1, 2, 6, 7] which also considered only sequence classification benchmarks for evaluation. Our experimental design was motivated by Chen et al., 2021 [1]. LRA benchmark [2] is a widely popular evaluation suite used for evaluating the efficiency of self-attention-based models. This benchmark evaluates the capabilities of different encoding models to capture  spatiotemporal dependencies within long sequences. We evaluate TransJect on a total of seven encoding tasks, where our model displays superior performances over the other Transformer variants. Our future work will involve evaluating our model on other large-scale language modeling and language understanding tasks in future.

---

> ### Author Response · Authors · 2023-04-11
> **Author Response to Reviewer 1wMf 2/2**
>
> ## (Comments Continued) ##
>
> **Question 1.** Why can cosine distance be greater than 1 in Figure 1?
>
> **Response.**  We use the implementation by Scikit-learn [9] that uses $1-cosine$ $similarity$ as the cosine distance. Cosine similarity is the cosine of the angle between two vectors which lies in $[-1,1]$. Therefore, the distance lies in $[0,2]$. We have now highlighted this   in Section B.2 of  our revised paper.
>
> **Question 2.** $\dots$  why preserving the Euclidean similarities is important?
>
> **Response.** Preserving cosine distance between different tokens ensures  the semantics (angle) is preserved. Preserving Euclidean distance ensures that they are not projected to distant subspaces. For example, $(0,1)$ and $(1,0)$ have a cosine distance of $1$ and a Euclidean distance of $1.41$. $(0,100)$ and $(100,0)$ have a cosine distance of $1$ and a Euclidean distance of $14.1$. Therefore, preserving both is important to ensure the tokens are projected similarly layer-wise.
>
> **Question 3.** $\dots$  how to enforce $W_1$ and $W_2$ to be orthogonally parameterized?
>
> **Response.**  We follow the orthogonal parameterization implementation by PyTorch [10]. This implementation uses different orthogonal maps (e.g., householder or Cayley mapping) to ensure orthogonality. We have now highlighted this information in Section B.2 of our revised manuscript.
>
> ### References
>
> [1] Chen, Yifan, Qi Zeng, Heng Ji, and Yun Yang. *Skyformer: Remodel self-attention with gaussian kernel and nystr\" om method.* Advances in Neural Information Processing Systems 34 (2021): 2122-2135.
>
> [2] Tay, Yi, Mostafa Dehghani, Samira Abnar, Yikang Shen, Dara Bahri, Philip Pham, Jinfeng Rao, Liu Yang, Sebastian Ruder, and Donald Metzler. *Long range arena: A benchmark for efficient transformers.* arXiv preprint arXiv:2011.04006 (2020).
>
> [3] Xiong, Yunyang, Zhanpeng Zeng, Rudrasis Chakraborty, Mingxing Tan, Glenn Fung, Yin Li, and Vikas Singh. *Nyströmformer: A nyström-based algorithm for approximating self-attention.* In Proceedings of the AAAI Conference on Artificial Intelligence, vol. 35, no. 16, pp. 14138-14148. 2021.
>
> [4] Zhu, Chen, Wei Ping, Chaowei Xiao, Mohammad Shoeybi, Tom Goldstein, Anima Anandkumar, and Bryan Catanzaro. *Long-short transformer: Efficient transformers for language and vision.* Advances in Neural Information Processing Systems 34 (2021): 17723-17736.
>
> [5] Chen, Beidi, Tri Dao, Eric Winsor, Zhao Song, Atri Rudra, and Christopher Ré. *Scatterbrain: Unifying sparse and low-rank attention.* Advances in Neural Information Processing Systems 34 (2021): 17413-17426.
>
> [6] Howard, Jeremy, and Sebastian Ruder. *Universal language model fine-tuning for text classification.* arXiv preprint arXiv:1801.06146 (2018).
>
> [7] Le, Hung, Truyen Tran, and Svetha Venkatesh. *Learning to Remember More with Less Memorization.* In International Conference on Learning Representations.
>
> [8] Kim, Hyunjik, George Papamakarios, and Andriy Mnih. *The Lipschitz constant of self-attention.* In International Conference on Machine Learning, pp. 5562-5571. PMLR, 2021.
>
> [9] https://scikit-learn.org/stable/modules/generated/sklearn.metrics.pairwise.cosine_distances.html
>
> [10] https://pytorch.org/docs/stable/generated/torch.nn.utils.parametrizations.orthogonal.html

---

### Review · Reviewer_3bzP · 2023-04-07

**Summary Of Contributions:**

- The authors propose a new parameterization of the Transformer, called TransJect, whose layers are claimed to be injective and also giving rise to lower entropy representations than standard Transformers.
- Some theory is given to highlight the contractive property of the newly proposed self-attention layer in TransJect, hence the injectivity of the residual map that uses the self-attention with a skip-connection.
- Experiments on short-range and long-range text classification tasks are given.
- Analysis of activation bounds and entropy of layer-wise representations are given for models trained on the short range classification task

**Audience:**

Yes

**Claims And Evidence:**

No

**Requested Changes:**

- Correct proof of the injectivity of the newly proposed residual self-attention map in Lemma 3.
- Clearer description and organization of the motivation behind wanting to design an injective Transformer.
- Empirical justification of using MoE rather than multihead self-attention
- Better justification of the choice of the baselines and benchmarks, or empirical results on more relevant baselines such as language modelling.


**Strengths And Weaknesses:**

**Strengths**
- a novel parameterisation of the Transformer
- Extensive analysis of activation factor and entropy of intermediate representation of resulting models

**Weaknesses**

Correctness
- I'm not convinced that the proof of Lemma 3, that the residual map is injective, is correct. The proof relies on F, the self-attention layer proposed in Theorem 1, being contractive, i.e. Lipschitz constant < 1. However Theorem 1 only shows that the activation bound of F ($sup_{X \neq 0} ||F(X)|| / ||X||$) is less than 1 (under the assumption that the largest eigenvalue of X^T X < 1), but there is no result about the Lipschitz constant of F ($sup_{X \neq Y} ||F(X)-F(Y)|| / ||X-Y||$). Note that these are different quantities, and in fact the Lipschitz constant is at least as big as the activation bound (by setting $Y=0$). An example of a function which has activation bound <= 1 but not Lipschitz (i.e. Lipschitz bound is infinite) is $F(x)=x sin(x)$. Its activation bound is clearly 1, but its derivative $F'(x)=sin(x) + x cos(x)$ is unbounded, so isn't Lipschitz. If you want an example on a bounded domain, say -$1 < x < 1$, $x \sqrt{1-x^2}$ has unbounded derivatives at the bounds but its activation bound is 1. This gap between activation bound and lipschitz constant is also present in the background section, where the authors suddenly jump from results on activation bounds to Lispchitz bounds. Unless I have misunderstood, this would be a major theoretical error since now there is no guarantee that the "Injective Transformer" is injective.

Motivation
- Also the assumption in Theorem 1, that the largest eigenvalue of $XX^T$ is 1, is going against the point of trying to design a Lipschitz version of self-attention. Such a constraint on the input space means that the input space is bounded, and then any continuous function is Lipschitz on a bounded space, including the original dot-product self-attention (see e.g. end of Section 3.1 in Kim et al. 2021).
- The motivation behind wanting to design an injective self-attention is not very clear. The introduction emphasizes the importance of preservation of semantic similarity and Euclidean distance across different layers of a Transformer. However it's not at all clear to me how injectivity is connected to such properties.
- The motivation for using the Mixture of Expert parameterization is not very clear to me - it seems quite orthogonal to the newly proposed self-attention, and there doesn't seem to be an ablation that uses the newly proposed self-attention with standard multi-head parameterization, so it's not clear whether the empirical improvement is coming from the Mixture of Expert or the newly proposed self-attention.

Clarity / Organization
- I'm also confused about what Figure 1 is trying to show. The results are quite different across the Transformer and BERT. Also how can cosine distance be > 1 in the plot?

Empirical Results
- While the proposed model seems to improve upon the baselines on the given benchmarks, it's not clear to me why these benchmarks & baselines were chosen. Firstly, why only look at classification tasks and not language modelling with perplexity as the metric? Transformers are mostly popular for language modelling tasks, and for long range classification tasks state-space models like S4 are much more dominant e.g. LRA SOTA: https://paperswithcode.com/sota/long-range-modeling-on-lra
- The given baseline results for the LRA benchmark are well below SOTA (in the above link), whose average score is in the 80s (as opposed to the 50s shown in the paper). None of the top 10 models in the link were compared against in the paper, even those that rely on attention rather than state-space models. Hence it makes me question the relevance of this paper. There is no explanation of this in the paper.

---

> ### Author Response · Authors · 2023-04-11
> **Author Response to Reviewer 3bzP 1/2**
>
> We thank you for the valuable feedback. We address the raised concerns. We have also highlighted the changes in our manuscript.
>
>  **W1.** Correctness
>
> **Response.** We apologize for the confusion. Let us clarify this here.
>
> Here $F$ is the nonlinear operator defined as $ELU(x U \Sigma V)$. Using Lemma 2, we can derive the Lipschitz bound for $ELU$ non-linear function as 1 (derivative of $ELU$ is bounded by 1). Being a linear mapping, both Lipschitz bound and activation bound are the same for $U \Sigma V$, *i.e.,* $||U \Sigma V||$. Using the orthogonality conditions of $U$ and $V$ and the fact that the largest eigenvalue of $\Sigma$ is 1, we derive the Lipschitz bound of $U \Sigma V$ as 1. Using the chain rule of differentiation and properties of supremum, we calculate $sup_{x} || \nabla_x F(x) || \leq sup_{x}|| \nabla_x ELU(x) || \cdot  || U \Sigma V|| = 1$. Therefore, the Lipschitz bound of $F$ is 1. Hope it resolves your confusion.
>
> We have also clarified this in the proof, described in Appendix A.4 of our revised manuscript.
>
> **W2.** Motivation
>
> **Response.** Sorry for the confusion again. Below, we clarify the points raised.
>
> * By enforcing the original embedding space compact, we can ensure that dot-product self-attention is Lipschitz, as explained by Kim et al., 2021 [1]. However, the authors suggested that the Lipschitz constant depends on the inputs. On the other hand, with simple non-normalized orthogonal parametrization (as shown in Section 4.1 of our paper), we reduce the non-linear attention to a simple linear transformation. The proposed attention operator is already Lipschitz continuous, bounded by the largest singular value of $X$. However, to ensure that the Lipschitz constant input is invariant, we enforce the largest singular value to be $1$.
>
> * Lipschitz condition ensures that the encoding layers do not project the tokens to an unbounded subspace (or to a subspace where the Euclidean distance is too high). However, it does not guarantee that the function is not contractive and does not squeeze the embeddings too much. We enforce an injectivity constraint in our proposed TransJect model to restrict this. As explained in Section 1 of our paper, injectivity is a regularization in our model. Therefore, our model does not require other implicit regularizations such as dropout.
>
> * The standard multi-head self-attention is not necessarily injective. To prove the injectivity of self-attention, we need to make sure that $W^{O}$ (defined in Section 3.2.2 of Vaswani et al., 2017 [2]) is full-rank. Instead, we use a convex combination of different attention heads (experts). Individually, each expert learns an injective mapping. Due to the convex combination, the mixture of experts (MoE) is also injective. We have now added this in corollary 2 of our revised manuscript and furnished the proof in Appendix A.5. We conduct a small-scale ablation (c.f. Figure 3) showing that TransJect with one expert can also outperform vanilla Transformer in IMDb and Char-IMDb classification tasks.
>
>  **W3.** Clarity / Organization
>
>  **Response.** Figure 1 shows that fine-tuned Transformer and pre-trained BERT learn different projections of the same tokens across different layers. Although the cosine distance between semantically relevant tokens decreases over layers, the Euclidean distance increases (at least for Transformer) over the layers, which suggests that the embeddings are projected onto a sparser subspace. We use the implementation by Scikit-learn [3] that uses $1-cosine$ $similarity$ as the cosine distance. Cosine similarity is the cosine of the angle between two vectors, which lies in $[-1,1]$. Therefore, the distance lies in $[0,2]$. We have now added this information in Section B.2 of our revised manuscript.  Note that in Figure 1, the results of Transformer and BERT are quite different as the former is trained from scratch on the IMDb classification task, whereas the latter is pre-trained on the language modeling task we obtained from Huggingface [4]. Section 1 of our paper mentions that sparser embeddings lead to higher entropy and inefficient encoding, particularly for long-range sequences.

---

> ### Author Response · Authors · 2023-04-11
> **Author Response to Reviewer 3bzP 2/2**
>
> ## (Comments Continued) ##
>
> **W4.** Empirical Results
>
>  **Response.** Our proposed  TransJect model is an injective parameterization of Transformer. Therefore, we only chose Transformer and its variants as the baselines for a fair comparison. Moreover, the motivation of our work is to ensure injectivity within self-attention and ensure analytical Lipschitz continuity in the Euclidean space. As discussed in Section 7 of our paper, TransJect depends on the eigenvalue decomposition of the matrix containing token embeddings, which makes it difficult to incorporate for tasks involving decoding (as context is limited during decoding means $XX^{T}$ is sparser). We evaluate our model on a total of seven encoding tasks, where TransJect displays superior performances over the other Transformer variants. We fail to evaluate our model on other language modeling tasks, primarily due to the lack of computational resources. Our future work will explore the proposed methodology for other language modeling and language understanding tasks.
>
>
> **Question 1.** Correct proof of the injectivity of the newly proposed residual self-attention map in Lemma 3.
>
> **Response.** Please see the response to W1.
>
> **Question 2.** Clearer description and organization of the motivation behind wanting to design an injective Transformer.
>
> **Response.** Please see the response to W2.
>
> **Question 3.** Empirical justification of using MoE rather than multi-head self-attention.
>
> **Response.** Please see the response to W2.
>
> **Question 4.** Better justification of the choice of the baselines and benchmarks or empirical results on more relevant baselines such as language modelling.
>
> **Response.** Please see the response to W4.
>
> ### References
>
> [1] Kim, Hyunjik, George Papamakarios, and Andriy Mnih. *The Lipschitz constant of self-attention.* In International Conference on Machine Learning, pp. 5562-5571. PMLR, 2021.
>
> [2] Vaswani, Ashish, Noam Shazeer, Niki Parmar, Jakob Uszkoreit, Llion Jones, Aidan N. Gomez, Łukasz Kaiser, and Illia Polosukhin. *Attention is all you need.* Advances in neural information processing systems 30 (2017).
>
> [3] https://scikit-learn.org/stable/modules/generated/sklearn.metrics.pairwise.cosine_distances.html
>
> [4] https://huggingface.co/models

---

> > ### Comment · Reviewer_3bzP · 2023-04-13
> > **Reviewer Response to Author rebuttal**
> >
> > W1
> > - You had originally defined Equation (4) $f(X) = XU \Sigma V$ from Equation (3), where $U=W^Q {W^K}^T \bar{Q}$, $V=\bar{Q}^TW^V$, and $X^T X = \bar{Q}\Sigma\bar{Q}^T$. This means that $\bar{Q}$ and $\Sigma$ are functions of $X$, hence $U, \Sigma, V$ are all functions of $X$. This implies that $f(X)$ is a non-linear function of $X$, so it's incorrect to say that $\sup_X ||f(X)|| = ||U\Sigma V||$, unless you are ignoring the dependence of $\bar{Q}$ and $\Sigma$ on $X$, and just treating $f(X) = XU \Sigma V$ as a linear map. But then in what sense is this self-attention? This is just a linear map. Also then what is the novelty of this method?
> >
> > W2
> > - Your first bullet point under W2 brings me back to the question above - if TransJect is simply replacing self-attention with a linear map (with orthogonal weight matrices) in a Transformer, in order to have a contractive residual mapping and hence an injective mapping, then it'd be difficult to even call this a Transformer as now there is no self-attention involved. If I understood correctly, I'd say this is very far from your original motivation of designing an injective Transformer.
> > - I think you're mixing up the terms "contractiveness" and "injectivity". If g(x) is a contractive function, then f(x) = x + g(x) is injective. But the properties such as preservation of semantic similarity and Euclidean distance are properties of contractive functions i.e. g, rather than f. Note that f can have Lipschitz constant > 1, so these quantities need not be preserved. So it's confusing that you're saying injectivity is responsible for such desirable properties, when it's actually the contractive property that is responsible. You can have an injective map such as f(x) = 100 * x + 1 that has neither of these properties.
> > - if W^O is parameterized as another learnable orthogonal map, wouldn't it be full rank?
> >
> > W3
> > - Thank you for the explanation on cosine distance. Re BERT, if the results are different from the Transformer since BERT has been trained on a language modelling task, then what is the value in adding these BERT results to Figure 1? Also does this mean that the issues you're trying to address with TransJect are only present for the text classification tasks and not necessarily for language modelling tasks?
> >
> > W4
> > - I don't understand what you mean by "TransJect depends on the eigenvalue decomposition of the matrix containing token embeddings, which makes it difficult to incorporate for tasks involving decoding (as context is limited during decoding means XX^T is sparser)" - are you saying that it is computationally expensive to compute the eigenvalues for every step of an autoregressive Transformer? If so this should be stated more clearly in the manuscript, that it is computationally infeasible to apply the approach to autoregressive Transformer based tasks such as language modelling, as this is a key limitation of the approach.

---

> > > ### Author Response · Authors · 2023-04-17
> > > **Follow Up Response to Reviewer 3bzP**
> > >
> > > **W1**
> > >
> > > As described in Section 4.1 of our paper, we calculate the reconstruction loss $||X^{T}X - U \Sigma U^{T}||$, which captures the non-linear dependence between different token embeddings. Once we calculate $U$ and $\Sigma$, we ignore the dependence of $X$ on $U$ and $\Sigma$ and thus simplify the attention operation with a linear mapping. Therefore, we capture attention only once at $l=1$, preserve the information in $\Sigma$ and $U$ and relearn the different linear projections across subsequent layers. This acts as an inductive bias to our proposed attention framework.
> > >
> > > **W2**
> > >
> > > * The reason we call the linear operator *attention* lies in the above response (response to W1). Through the reconstruction loss, our model captures the inter-dependence between different tokens. The eigenvectors $U$ and eigenvalues of $\Sigma$ effectively preserve this information and propagate it to our proposed attention module. Hence, albeit being a linear operator, each token attends to the other in our attention module through the minimization of reconstruction loss.
> > >
> > > * We apologize for the confusion. Note that we are using the properties of contraction of $\frac{\alpha}{L}F(X)$ to prove that the residual mapping $X + \frac{\alpha}{L}F(X)$ is injective. Also, note that the residual map has a Lipschitz bound $> 1$. We never claim that this is a contraction mapping. In this work, we design a self-attention-based encoder model to preserve the Euclidean distance up to a theoretical bound $e^{2}$, not necessarily a contraction. Therefore, we conclude that the Lipschitz continuity (bound > 1, not contraction) is responsible for preserving the distance within an analytical bound and attaining lower entropy. On the other hand, injectivity is accountable for the efficiency of our model and for making it scalable at large depths.
> > >
> > > * Yes, with orthogonal parameterization, one can make the multi-head self-attention injective. However, the residual $X + Attn(X)$ would still not be injective. One possible way to make it injective is by enforcing $Attn(X)$ as a contraction mapping.
> > >
> > > **W3**
> > >
> > > Although the activation bound for BERT in Euclidean space is lower than that of the Transformer, we can still observe a non-monotonous pattern for BERT in Figure 1. For example, the semantic similarity between the words "good'' and "effect'' drops after layer $4$ and then increases again after layer $9$. On the other hand, the Euclidean distance between the same pair starts increasing after layer $2$ and then decreases after layer $7$. As part of our motivation, we claim that the distance between two semantically relevant tokens should follow some consistent trend over the encoding layers to ensure that the model preserves the layer-wise information propagation. Therefore, although having a robust training objective like language modeling can lower the activation bound of the encoder models, it still may not be sufficient to preserve layer-wise consistency of distance preservation. Note that there does not exist any analytical (data-independent) Lipschitz bound for BERT or vanilla Transformer, under standard conditions. In this paper, using the Lipschitz properties, we calculate the Lipschitz bound of our model $e^2$.
> > >
> > > **W4**
> > >
> > > As Section 7 of our paper mentions, having a sparser $X^{T}X$ can lead to more information loss during the reconstruction of $X^{T}X$ using the pseudo eigenvalues and eigenvectors. It is not computationally expensive but less effective in preserving the inter-token dependencies. We also argue in Section 5.2 that the same reasoning can also be used to justify the less effective performance of our model on the AGNews classification task, where the context was limited. When the context is limited, Random-TransJect performs much better, as it uses initialized eigenvalues $\Sigma$ randomly and does not depend on the $X^{T}X$. Our work primarily explores the relationships between efficiency, model entropy and activation. We will use these findings to devise efficient auto-regressive (decoder) models in our future works.

---

> > > > ### Comment · Reviewer_3bzP · 2023-04-20
> > > > **Reviewer response to reviewers**
> > > >
> > > > W1
> > > > - I think this exposes an inconsistency between your methodology in practice and the theory you propose. So in practice your $U$ & $V$ & $\Sigma$ in $f(X)=XU \Sigma V$ are computed using X, but for the theory you ignore the dependence on X when you derive the Lipschitz bound on $f$. This inconsistency hurts the relevance of the theory. i.e. you have derived the Lipschitz bound of $f_{theory}(X)=XU\Sigma V$ under the assumption that $U, \Sigma, V$ are independent of $X$. However this does NOT give us a Lipschitz bound of the computation you apply in practice, $f_{practice}(X)=X U(X) \Sigma(X) V(X)$. The same holds for the claims around injectivity - the theory doesn't tell us whether the residual mapping used in practice is actually injective or not. I believe this is a significant limitation for the theory of this paper.
> > > >
> > > > W2
> > > > - Thank you for the clarifications. I think adding these clarifications to the paper would help its presentation.
> > > >
> > > > W3, W4
> > > > - I'm still concerned about the practical applicability of TransJect at scale, if you have to minimize the reconstruction loss $||X^T X - U\Sigma U||$ wrt $U,\Sigma$ for each forward pass of the model, even if it's only applied for the first layer. For computational efficiency I imagine you'd need to make the optimization suboptimal, which can give poor reconstruction accuracy and thus a lower performance. Also at this point the learned $U, \Sigma$ are far from the true eigendecomposition of $X^TX$, so the practice deviates even further from the theory.

---

> > > > > ### Author Response · Authors · 2023-04-21
> > > > > **Follow Up Response to Reviewer 3bzP**
> > > > >
> > > > >
> > > > > **W1**
> > > > >
> > > > > We apologize for the confusion. Note that even in practice, we resort to approximated eigenvalue decomposition. Although we use eigenvalue decomposition as part of the motivation, the Lipschitz bound of $Attn(X)$ does not depend on the eigenvalue decomposition. It only depends on the largest singular value of $\Sigma$ and the Lipschitz bound of $U$ and $V$ if we assume that $X^{T}X$ can be approximated with $\tilde{Q}\Sigma \tilde{Q}^{T}$. The injectivity of the residual also does not depend on the exact eigenvalue decomposition; rather it depends on the contraction properties of $\frac{\alpha}{L}ELU(Attn(X))$. Although having exact eigenvalues $\Sigma$ and eigenvectors $\tilde{Q}$ of the matrix $X^{T}X$ ensure that the Lipschitz bound of $Attn(X)$ is 1, it is not a necessary condition. Therefore, for practical reasons, we chose approximated eigenvalue decomposition (highlighted in Section 4.1). Further, by assuming $U$, $V$ and $\Sigma$ to be learnable parameters (property of the model rather than dependent on $X$), where $U$ and $V$ are orthogonally parameterized, and $\Sigma$ is min-max scaled normalized, we ensure that the Lipschitz bound of the operation $ELU(XU \Sigma V)$ is 1. By minimizing the reconstruction loss, we ensure that the learned $U$ and $\Sigma$ encode the token-wise similarities (calculated using $X^{T}X$), so we can still treat this as an attention operation.
> > > > >
> > > > > **W2**
> > > > >
> > > > > We are happy to be able to resolve your confusion. We will also add these clarifications in the main manuscript.
> > > > >
> > > > > **W3, W4**
> > > > >
> > > > > As explained, to make the theory work, we do not need to compute the exact decomposition of $X^{T}X$. We only need to ensure that the learned parameters $U$ and $\Sigma$ have a Lipschitz bound of 1. As explained in response to W1, by assuming $U$, $V$ and $\Sigma$ to be learnable parameters (property of the model rather than dependent on $X$), where $U$ and $V$ are orthogonally parameterized, and $\Sigma$ is min-max scaled normalized, we ensure that the Lipschitz bound of our proposed attention operation $XU \Sigma V$ is 1.

---

> > > > > > ### Comment · Reviewer_3bzP · 2023-04-28
> > > > > > **Reviewer response to authors**
> > > > > >
> > > > > > W1
> > > > > > I understand that in practice you resort to an approximation of the eigenvalue decomposition, but I don't think this resolves the issue I'm pointing to. The very fact that you're using approximations $\tilde{U}$ , $\tilde{\Sigma}$ that are the result of optimizing $||X^TX-U\Sigma U^T||$ wrt $U$ and $\Sigma$ means that your approximations $\tilde{U}$ , $\tilde{\Sigma}$ are dependent on $X$, i.e. they are functions of $X$. So then pretending that these values are now independent of $X$ for the purpose of the theory when you derive Lipschitz bounds is inconsistent with the practice, where the optimization procedure used to obtain $\tilde{U}$ , $\tilde{\Sigma}$ are in fact dependent on $X$.
> > > > > >
> > > > > > W3, W4
> > > > > > The question was about scalability/computational efficiency of the proposed TransJect. It seems like there's a tradeoff between computation and performance, and it's not clear whether scalable versions of TransJect (by reducing computation) will be performant enough for tasks at scale, especially for tasks such as language modelling which are not explored in the paper nor the rebuttal.

---

> > > > > > > ### Author Response · Authors · 2023-05-03
> > > > > > > **Follow Up Response to Reviewer 3bzP**
> > > > > > >
> > > > > > > **W1**
> > > > > > >
> > > > > > > As discussed previously, we resort to simplifying attention operation by considering the dependence of $X$ on $\tilde{U}$ and $\tilde{\Sigma}$ only in the reconstruction loss, not during the attention projection, as the approximated eigenvalue and eigenvectors are used across layers and do not depend of $X^{(l)}$ for $l \geq 1$. This simplification helps us to compute the Lipschitz bound of constrained non-normalized linear orthogonal attention. In our future work, we plan to analytically calculate the Lipschitz bound of the non-linear orthogonal attention $XWX^{T}XW^{V}$ without eigen approximation. With this, we can compute a tighter activation bound for linear non-normalized self-attention.
> > > > > > >
> > > > > > > **W3,W4**
> > > > > > >
> > > > > > > As discussed previously, we present a pioneering attempt to explore a low entropic self-attention-based model for long and short-range classification tasks in this work. The superior results achieved by TransJect on the LRA benchmark and short sequence classification tasks encourage us to explore the model for other encoding and language modelling tasks. Also, Table 3 in our paper highlights the efficiency of TransJect on a long-range encoding task. Being a low-entropic model, TransJect is efficient in encoding very long sequences. The primary limitation of our model is its inability to encode sparse sequences. In future, we plan to overcome this sparsity limitation by using the non-approximated non-normalized orthogonal attention instead of the eigen approximation.  Our future work will involve evaluating our model on other large-scale language modelling and language understanding tasks.

---

### Decision · Action_Editors · 2023-05-08

**Recommendation:** Reject

**Comment:**

The submission investigates a Transformer encoder model called TransJect which uses a Lipschitz-continuous alternative to dot-product attention—and is claimed to learn injective mappings that preserve the layer-wise Euclidean distances between token pairs as a result—along with additional architectural modifications. Theory is presented to support claims made about the proposed approach's properties, and empirical evaluations are reported on movie sentiment and topic classification benchmarks.

Reviewers found the proposed idea and the phenomenon it attempts to control interesting, but following discussion with the authors they remain concerned with the submission's empirical and theoretical support and recommend rejection.

**Audience:**

The submission if of interest to TMLR's audience. Reviewers expressed interest in the problem of bounding pairwise distances across Transformer layers.

**Claims And Evidence:**

Following the discussion with the authors, reviewers remain concerned with the following points:
- There is a gap between the theory presented and the way the proposed approach is applied in practice. Reviewer 3bzP questions the validity of the derived Lipschitz bound given TransJect's implementation.
- Empirical support needs improvement. Reviewer D4jj remains unconvinced given the evidence presented that Transformers are limited in their performance by their inability to preserve embedding distances. Reviewers 1wMf and 3bzP remain concerned that there are no results on language modelling. Reviewer 3bzP points out that a naive implementation of TransJect is computationally demanding, but that the computation-performance tradeoff made in practice is insufficiently discussed.